# Nigrostriatal dopamine pathway regulates auditory discrimination behavior

Allen P. F. Chen[1,2], Jeffrey M. Malgady [1], Lu Chen[1], Kaiyo W. Shi [1],
Eileen Cheng[1,3], Joshua L. Plotkin [1,4], Shaoyu Ge[1] & Qiaojie Xiong [1]✉

The auditory striatum, the tail portion of dorsal striatum in basal ganglia, is implicated in perceptual decision-making, transforming auditory stimuli to action outcomes. Despite its known connections to diverse neurological conditions, the dopaminergic modulation of sensory striatal neuronal activity and its behavioral influences remain unknown. We demonstrated that the optogenetic inhibition of dopaminergic projections from the substantia nigra pars compacta to the auditory striatum specifically impairs mouse choice performance but not movement in an auditory frequency discrimination task. In vivo dopamine and calcium imaging in freely behaving mice revealed that this dopaminergic projection modulates striatal tone representations, and tone-evoked striatal dopamine release inversely correlated with the evidence strength of tones. Optogenetic inhibition of D1-receptor expressing neurons and pharmacological inhibition of D1 receptors in the auditory striatum dampened choice performance accuracy. Our study uncovers a phasic mechanism within the nigrostriatal system that regulates auditory decisions by modulating ongoing auditory perception.

Auditory information is crucial for daily life experience, informing an animal's decisions and survival. How neural pathways transform auditory stimuli into salient actions and decisions is a topic of intense study[1–3]. Differential sensory encoding and cortical mechanisms have previously been proposed to underlie basic auditory decision-making and discrimination processes[4–9]. Neuromodulatory pathways, which are evolutionarily ancient subcortical systems integral to motivation and learning, have emerged as critical regulators of auditory neural circuits that govern complex auditory behaviors, such as maternal care, vocal communications, and reinforcement feedback[10–15]. However, how neuromodulatory motifs and neural circuits jointly facilitate simple auditory-guided decisions remains unclear.

The auditory striatum, a caudal tail portion of the dorsal striatum that receives auditory pathway inputs, is uniquely positioned within the basal ganglia to integrate ongoing auditory stimuli and enforce action choices[16–21]. This striatal region receives dopaminergic inputs from a distinct population of substantia nigra pars compacta (SNc)

neurons[22] that have previously been implicated in aversive threat reinforcement and salience[23,24]. Consistently, auditory corticostriatal activity has been implicated in encoding behavioral responses to threatening auditory stimuli[25]. However, the mechanism through which dopamine influences the basic activity of the auditory striatum and the role of dopamine modulation in auditory decision-making remain unknown.

Midbrain dopamine signaling has been broadly studied in the field of reinforcement learning as a teaching signal that encodes reward prediction errors[26–30]. Mounting evidence also revealed functional heterogeneity within the midbrain dopamine system. Midbrain dopaminergic projections to different locations have been found to encode time perception, stimulation difficulty, action, confidence, locomotion, movement, and aversive outcomes[23,31–39]. However, whether and how dopamine influences sensory perception during concurrent decisions remains unknown, particularly in the context of auditory-cued behaviors involving the tail striatum. Few studies exploring the

[1]Department of Neurobiology and Behavior, Stony Brook University, Stony Brook, NY 11794, USA. [2]Medical Scientist Training Program, Renaissance School of Medicine at Stony Brook University, Stony Brook, NY 11794, USA. [3]Department of Physiology and Biophysics, Stony Brook University, Stony Brook, NY 11794, USA. [4]Center for Nervous System Disorders, Stony Brook University, Stony Brook, NY 11794, USA. ✉e-mail: qiaojie.xiong@stonybrook.edu

role of dopamine in sensorimotor decision-making have considered the topographically segregated regions of the striatum[40–43]. Here, we investigated how dopamine modulates the auditory striatum during auditory perception and decision-making.

To address the role of auditory striatal dopamine regulation in decision-making, we used a combination of rodent behavioral assays, in vivo imaging, optogenetic/chemogenetic, anatomical methods, and pharmacological approaches. Our study identified a specialized nigrostriatal locus involved in perceptual performance that is recruited and presents with amplified activity in response to increased stimulus discrimination difficulty.

## Results

### The auditory nigrostriatal pathway is critical for auditory discrimination behavior

Dopamine dysfunction has been linked to decision-making deficits in a variety of mental disorders, such as Parkinson's disease[44–48], suggesting a role for dopaminergic signaling in cognition. However, the underlying neural circuitry and regulatory dynamics remain poorly understood. In this study, using in vivo dopamine and $Ca^{2+}$ imaging methods, together with optogenetic manipulation, we explored the role of dopamine in an auditory behavior involving the striatal circuitry.

We first established an auditory frequency discrimination task utilizing a cue featuring overlapping pure tones with different frequencies, referred to as a 'cloud of tones,' as illustrated in Fig. 1a and previously described[20]. Mice were trained to engage in a tone-cued, two-alternative forced-choice task for water consumption. Briefly, water-restricted adult mice used center poking to self-initiate trials and learned to wait in the center port for a cue presentation (Fig. 1a, right panel). Mice were trained to report to the left- or right-side port in response to low- or high-frequency tones, respectively, to receive a water reward. After 3–5 weeks of training, mice were able to readily discriminate between high- and low-frequency tone mixtures and developed psychometric performance curves (Fig. 1b), as previously reported[16,19,20].

To examine the role of striatal dopamine in auditory discrimination behavior, we first identified dopaminergic inputs to the auditory striatum using a retrograde viral approach (Fig. 1c). The retrograde-capable canine adenovirus type 2[49] expressing Cre recombinase (CAV-Cre) was stereotaxically injected into the auditory striatum of Ai14[50] transgenic mice. Neurons projecting to the auditory striatum were retrogradely infected by CAV and expressed Cre recombinase, activating tdTomato expression. Two weeks after CAV-Cre injection, we euthanized the mice and analyzed the Cre-induced tdTomato signal across brain regions to identify midbrain dopaminergic inputs (Fig. 1c, d). Clusters of tdTomato+ neurons were found primarily in the SNc (Fig. 1d, top; $n = 4$ mice, 95.25 ± 1.48%) and were mostly tyrosine hydroxylase positive (TH+, 94.51 ± 1.12%), indicating their dopaminergic nature (Fig. 1d, bottom). For convenience, the SNc projection to the auditory striatum is referred to as the 'auditory nigrostriatal pathway' throughout the remaining text.

To determine whether the auditory nigrostriatal pathway is actively involved in auditory discrimination behavior, we assessed task performance following the optogenetic silencing of this pathway. As illustrated in Fig. 1e, this pathway was specifically targeted in a cohort of well-trained mice by injecting CAV-Cre into the auditory striatum and injecting an adeno-associated virus (AAV) expressing the Cre-dependent neuron silencer archaerhodopsin T (ArchT[51]), into the SNc. We bilaterally implanted optic fibers 400 μm above the SNc and tested task performance 3–4 weeks after surgery. During tone presentation, we delivered orange light pulses (530 nm) through the optic fiber to silence ArchT-expressing neurons in a small fraction (5%–10%) of randomly selected trials. To achieve sufficient silencing during tone presentation, as previously described[20], we delivered optical stimulation during both the pre-tone and tone phases (Fig. 1e). To control for the

potential influence of visual stimulation effects, during all trials, we included a masking light stimulation using the same presentation time window as silencing trials delivered through a 530-nm bulb placed above the center port. After all tests, mice were euthanized and analyzed to confirm correct ArchT expression patterns and optic fiber implantation sites (Supplementary Fig. 1a). In a separate group of mice, we also confirmed that light stimulations silenced ArchT-positive SNc neurons using whole-cell patch recording on acute brain slices (Supplementary Fig. 1b). Comparing task performance between optic fiber stimulation trials and masking light alone trials showed that the bilateral inhibition of the auditory nigrostriatal pathway during tone presentation resulted in impaired task performance (Fig. 1f; $p < 0.001$, two-sided Wilcoxon rank-sum test). No consistent directional bias was induced by optogenetic inhibition (Supplementary Fig. 1c, $p = 0.19$, two-sided Wilcoxon rank-sum test). To exclude effects on locomotion that might affect task performance, we analyzed other associated behavioral parameters. Optical silencing did not change subsequent movement times or early withdrawal rates during task performance (Supplementary Fig. 1d), suggesting that impaired performance was specific to cued decision-making without locomotor effects. To further validate this finding, we performed optical stimulation in a GFP-only set of animals and observed no effects on task performance (Supplementary Fig. 1e–h; **g**: $p = 0.98$, **h**: $p = 0.54$).

Next, we analyzed the impact of silencing the auditory nigrostriatal pathway on subsequent trials. Although auditory nigrostriatal silencing impaired the performance of the trial in which it was applied (current trial), subsequent trial performance remained intact in the absence of inhibitory signal (Fig. 1g, $p = 0.42$). Furthermore, silencing the auditory nigrostriatal pathway during the post-tone period did not influence discriminatory performance (Fig. 1h, i, $p = 0.54$) or movement parameters, such as movement time or early withdrawal rate (Fig. 1j, $p > 0.05$).

A SNc neuron that projects to the auditory striatum may also project to other brain regions or function to regulate local circuit activity. To exclude potential indirect effects, we performed dopaminergic terminal inhibition in the auditory striatum by injecting AAV-DIO-ArchT into the SNc of *DAT*-Cre mice[52] to restrict ArchT expression to SNc dopaminergic neurons. We then implanted optic fibers bilaterally above the auditory striatum (Supplementary Fig. 1i–l; post hoc validated, Supplementary Fig. 1m). In these mice, we observed similar task performance impairments following optical inhibition of dopaminergic terminals in the auditory striatum (Supplementary Fig. 1i–l; **k**: $p < 0.001$, **l**: $p = 0.09$, two-sided Wilcoxon rank-sum test), suggesting that impaired discrimination is likely due to the direct inhibition of dopaminergic projections to the auditory striatum. Our findings revealed that the auditory nigrostriatal pathway plays an essential role in modulating auditory discrimination behavior.

### Nigrostriatal dopamine modulates striatal auditory cue representations

Mice task performance was only affected in those trials in which optogenetic manipulations occurred simultaneously with tone presentations (Fig. 1e–i), suggesting a potential modulatory role for dopamine in auditory processing in the auditory striatum. We next explored whether and how nigrostriatal activity modulates striatal tone-evoked responses.

To monitor striatal neuronal activity in mice during manipulations of auditory nigrostriatal pathway, we performed simultaneous microendoscopic recording and chemogenetic inhibition, as previously described[53]. We injected AAV expressing the calcium indicator GCaMP6f[54] into the left auditory striatum, followed by gradient-index (GRIN) lens implantation above the auditory striatum, as illustrated in Fig. 2a. We also injected AAV expressing Cre-dependent hM4Di-mCherry[55] into the left SNc of a cohort of well-trained *DAT*-Cre mice for chemogenetic inhibition. After 3 weeks of recovery, we recorded $Ca^{2+}$

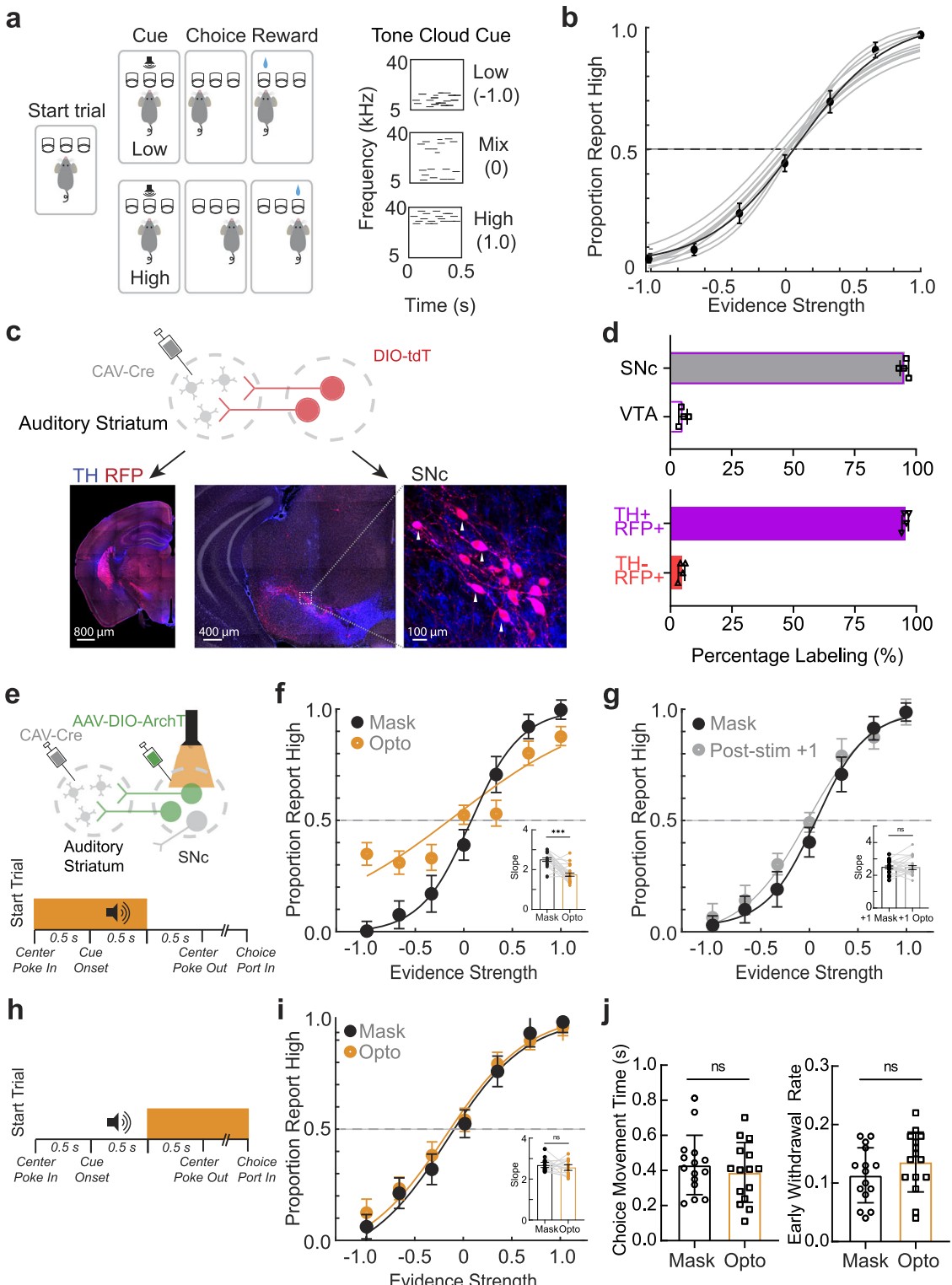

signals from auditory striatal neurons while chemogenetically suppressing the SNc via the systematic injection of clozapine N-oxide (CNO; Fig. 2a, b). Proper lens placement, local GCaMP6f striatal neuron expression, and local hM4Di-mCherry expression in dopaminergic terminals were verified in these mice after the behavioral experiments (Supplementary Fig. 2a). Based on previous studies[6,19,20], we expected that auditory striatal neurons would be active during the task, and we identified a subset of auditory striatal neuronal regions of interest (ROIs) with robust tone-evoked responses (72 out of 395 detected neuronal ROI, $p < 0.05$ Wilcoxon rank-sum test; Fig. 2c). The

proportion of tone-responsive neurons varied across animals (from 7% to 36%), mainly correlated with the field-of-view based on post-hoc validation. The more posterior striatum in the field of view, the higher portion of tone-responsive neurons we observed. In addition to tone-responsive neurons, we also identified a small fraction of neurons that responded to the white noise (Methods) sound in early withdraw trials (14 neurons, 9 overlapped with the tone-responsive ones), reward (19 neurons), and no reward outcome (6 neurons) (Fig. 2c, right panel). Although in this study we only examined well-trained mice, similar tone-evoked responses were found in naïve mice (Supplementary

**Fig. 1 | Silencing auditory striatal–projecting SNc neurons impairs auditory discrimination. a** schematic of the cloud-of-tones task. with representative stimuli of maximal evidence strength (±1) and minimal evidence strength (0). **b** Example psychometric curves ($n = 5$ animals, 10 sessions) and overlayed average. Evidence strength = (number of high tones − number of low tones)/(number of high tones + number of low tones). **c** Canine adenovirus type 2 expressing Cre (CAV-Cre) retrograde viral strategy for targeting auditory striatal–projecting substantia nigra pars compacta (SNc) neurons. Cartoon schematic and confocal imaging of the injection site and retrograde site demonstrating the labeling of SNc neurons. **d** Top, percentage of tyrosine hydroxylase (TH)$^+$ neurons located in the SNc vs. the ventral tegmentum area (VTA). Bottom, percentage of labeled neurons expressing TH ($n = 4$ mice). **e** Schematic of optogenetic silencing strategy. (Psychometric performance for tone–based silencing comparing masking light trials with optogenetic light trials. ($n = 5$ mice across 21 sessions). Inset, comparison of corresponding regression slopes for individual sessions. Bars are the means, and dots are the individual sessions ($^{***}p = 0.0009$, two-sided Wilcoxon rank-sum test). **g** Psychometric performance for tone–based silencing comparing masking light trials with trials immediately following optogenetic inhibition. ($n = 5$ mice across 21 sessions). Inset, comparison of corresponding regression slopes for individual sessions. Bars represent the mean, and dots represent individual sessions (n.s. $p = 0.42$, two-sided Wilcoxon rank-sum test). **h** Schematic for optogenetic post-tone silencing. **i** Psychometric performance for post-tone silencing comparing the masking light trials with post-tone inhibition trials. ($n = 5$ mice across 16 sessions). Inset, comparison of corresponding regression slopes for individual sessions. Bars represent the mean, and dots represent individual sessions (n.s. $p = 0.54$, two-sided Wilcoxon rank-sum test). **j** Post-tone silencing impact on choice movement time (n.s. $p = 0.19$, paired t-test) and early withdrawal rate ($p = 0.82$, paired t-test). Individual dots represent separate sessions across animals ($n = 5$ mice across 15 sessions). Data in **d**, **f**, **I**, **j** are presented as mean ± SEM. Source data are provided as a Source Data file.

Fig. 2b), consistent with previous report in rats[18]. Furthermore, these striatal responses showed preferences to tone frequencies but not movement directions (Supplementary Fig. 2c, d)

By registering neuronal ROIs across vehicle- and CNO-treated sessions and tracking changes in ROI activity[56], we found that CNO-mediated inhibition of SNc dopaminergic neurons significantly decreased tone-evoked activity in striatal neurons (Fig. 2d). The tone-evoked peak fluorescence of the neuronal ROI decreased significantly (Fig. 2e, $n = 21$ neurons from 3 hM4Di-expressing mice, $p < 0.001$, Mann–Whitney U test) in CNO-treated sessions relative to vehicle-treated sessions. Using a mCherry-only vector as control, we found that CNO treatment alone did not modulate auditory striatal neuronal activity ($n = 21$ neurons from 3 mCherry-expressing mice, $p = 0,67$, Fig. 2e). There is a significant difference between the changes in tonal response induced by CNO delivery in mCherry control vs hM4Di mice ($p < 0.05$, Mann Whitney U test).

The mice in our task were subjected to tones of varying frequencies and tone mixtures; therefore, we analyzed how SNc silencing impacted neuronal tone responses across different stimuli. We found that neuronal ROIs that were differentially activated by high-frequency tones, low-frequency tones, and tones associated with different evidence strengths (evidence strength = (number of high tones − number of low tones)/(number of high tones + number of low tones)) were all impacted by CNO treatment (Fig. 2f–i).

Interestingly, striatal neurons had a greater response towards task stimuli corresponding to the most ambiguous stimuli, at zero evidence strength (Fig. 2j; vehicle control injection, $p < 0.01$, Mann–Whitney U test). However, this greater response towards ambiguous stimuli diminished upon nigrostriatal silencing (Fig. 2k; CNO injection, $p = 0.80$, Mann–Whitney U test). We were able to detect 16 (Fig. 2l) out of 42 (Fig. 2j) neurons from both Vehicle and CNO sessions. Consistently, nigrostriatal silencing induced a greater reduction in tonal responses towards zero- than one-evidence strength stimuli (Fig. 2l, $p < 0.05$, Mann–Whitney U test).

These data indicate that nigrostriatal dopamine activity modulates auditory striatal tone representations during decision-making, and thus suggests a potential physiological mechanism for dopamine's impact on auditory perception and discrimination behavior. Furthermore, nigrostriatal activity appears to be important for evidence strength or difficulty in auditory striatal sound encoding.

## Nigrostriatal dopamine release is difficulty-dependent in the auditory discrimination task

The impairment of both task performance and striatal tone responses following SNc inhibition suggested that auditory nigrostriatal dopamine release may be responsive to tone presentations in task-performing mice. In addition, based on our characterization of evidence strength-dependent responses in the auditory striatum (Fig. 2j–l), we hypothesized that auditory nigrostriatal dopamine

activity may fluctuate depending on tonal evidence strength. To test this hypothesis, we established a microendoscopic approach for measuring striatal dopamine fluctuations in mice during task performance using a newly developed dopamine fluorescence sensor (the G-protein-coupled receptor activation–based series DA2m[57]; Fig. 3a, upper panel). In brief, we injected AAV expressing DA2m into the left auditory striatum and implanted a GRIN lens above the auditory striatum (Fig. 3a, lower panel) in a cohort of well-trained wild-type mice. Four weeks after surgery, we performed continuous recordings of fluorescent signals to study dopamine dynamics in the auditory striatum across 25–40 task sessions. Due to the ubiquitous membranous expression of this DA2m sensor, we analyzed the whole field-of-view dopamine dynamics within the auditory striatum (Supplementary Movie 1). To explore task performance–related changes in dopamine concentrations, we aligned recorded striatal dopamine activity with tone onset across trials, as shown in representative images and traces in Fig. 3b. As hypothesized, we found a reliable tone-induced dopamine release in the auditory striatum (Fig. 3c).

To exclude general fluctuations in fluorescence due to motion artifacts, we used the same recording and analysis approach in mice expressing GFP instead of DA2m and observed no changes in fluorescence (Supplementary Fig. 3a). To verify that this phasic response represents physiological changes in dopamine, we took advantage of the structural quenching of DA2m by D2 dopamine receptor (D2R)-specific antagonists[57]. We first adopted a reported in vivo protocol[57], (also see in Methods) and found that intraperitoneal (i.p.) injection of D2R-specific antagonist eticlopride (1.0 mg/kg) blunted tone-evoked DA2m responses in mice performing the task (Supplementary Fig. 3b, c, $p < 0.0001$). We next used acute brain slices to further verify the DA2m in our system. The external application of another D2R-specific antagonist, haloperidol (10 μM), diminished the increase in DA2m fluorescence evoked by SNc axon terminal activation using channelrhodopsin-2 (ChR2[58]), (Supplementary Fig. 3d). These data suggest that the tone-evoked increase in DA2m fluorescence observed in Fig. 3b, c is due to local changes in dopamine concentrations during task performance.

We next asked whether increasing task difficulty (using a mixture of low- and high-frequency tones) influences dopamine dynamics in the auditory striatum by examining tone-evoked dopamine release across individual trials. In the cloud-of-tones task, tones with different evidence strengths (0 to ±1) were randomly assigned to trials within a session, with ±1 representing the easiest trials (all low-frequency [−1] or all high-frequency [+1] tones) and 0 representing the hardest trials (equal numbers of low- and high-frequency tones). In this paradigm, well-trained mice chose the correct reward side more frequently during easy trials and performed at a random chance level during difficult trials (evidence strength = 0; Fig. 1b). We found that tones carrying 0 evidence strength induced the strongest increase in dopamine signals (Fig. 3d, e, $p < 0.0001$), suggesting that the tone-evoked dopamine

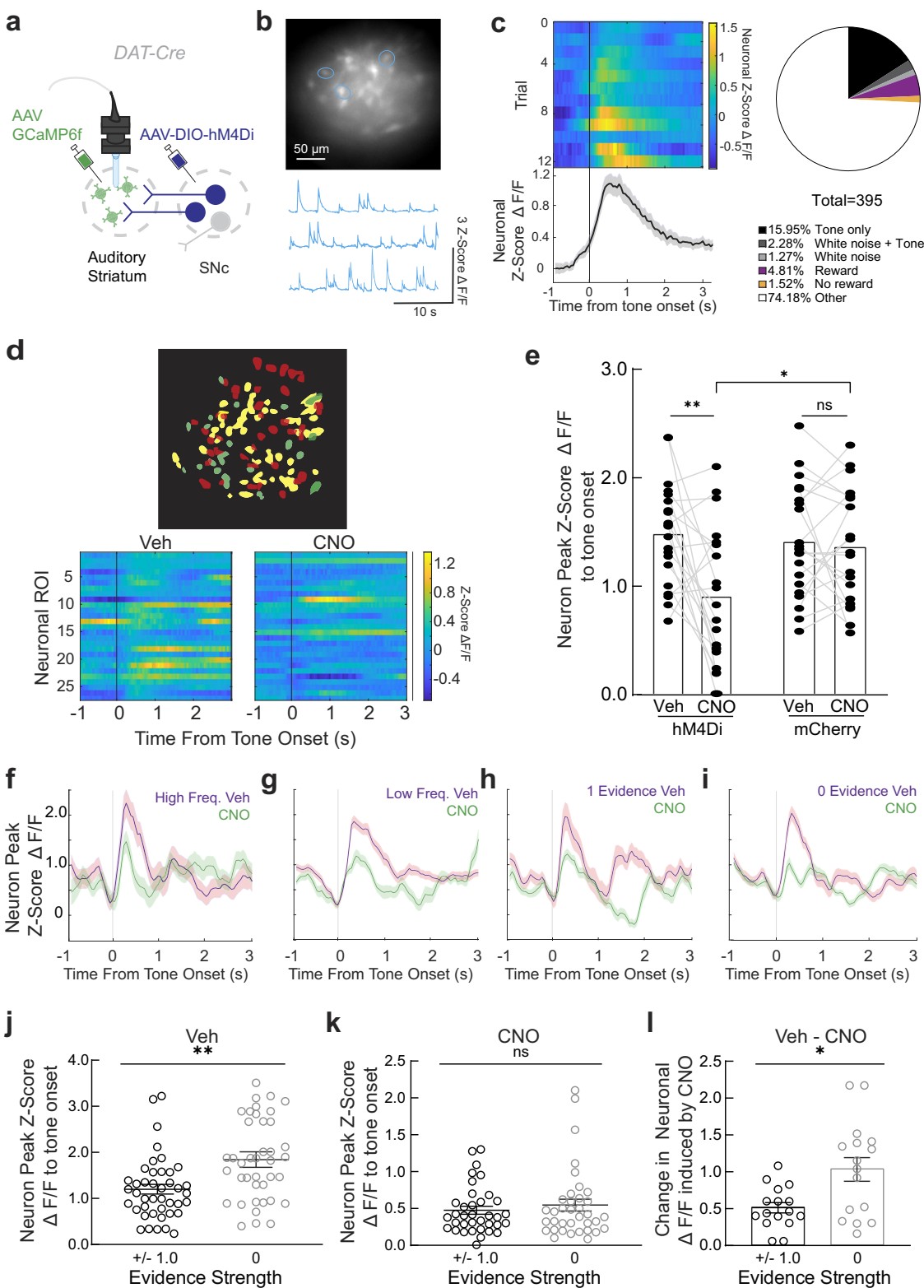

increase is positively correlated with task difficulty. This difficulty-dependent change occurs for both low- and high-frequency trials that direct leftward and rightward choices, respectively (Supplementary Fig. 3e), and similar changes in dopamine release were observed for both correct and incorrect trials (Supplementary Fig. 3f, g).

We did not observe any differences when comparing the dopamine responses induced by the receipt of the water reward across trials associated with the presentation of tones of different evidence strengths (Fig. 3f, g, **g:** $p > 0.05$), whereas the outcome-induced dopamine responses tended to be greater for correct trials than for incorrect trials (Supplementary Fig. 3h, i).

To exclude the possibility that this observed difficulty-dependent dopamine response was due to tone preference, we performed the same DA2m recording in a cohort of mice that passively listened to the same auditory stimuli, using the same duration and mixture of tones. Although tone-evoked dopamine responses were detected, the signal

**Fig. 2 | Silencing of the SNc impairs striatal tone responses during auditory discrimination. a** Schematic for simultaneous microendoscopic imaging of the auditory striatum and nigrostriatal chemogenetic inhibition. **b** Top, representative field-of-view imaging the auditory striatum. Bottom, representative corresponding detected traces. **c** Left, heatmaps and averaged trace of an example tone-responsive neuronal ROI aligned to tone onset (data are mean ± SEM). Right, Pie chart depicting the distribution of responsive neuronal ROIs. ROIs with significantly ($p < 0.05$) higher response profiles following different task variables: tones, white noise (in early withdrawal trials), reward (incorrect trials), or no reward (in error trials) are quantified. 'Other' refers to neurons with no significant responses to task variables. **d** Top, cellular registration and activity maps of an example ROI when administering vehicle (red) or CNO (green) prior to the auditory task. Bottom, individual change in mean tone-evoked response per neuronal ROI in vehicle and CNO contexts. **e** Peak trace response across registered neuronal ROIs for the vehicle (red) and CNO (green) contexts. Bars are mean and individual dots represent values from each neuron. $n = 21$ neurons from 3 mice from each group *$p = 0.0157$, **$p = 0.0032$, n.s. $p = 0.7652$, two-sided Mann–Whitney U test. **f–i** Averaged tone trace response of example neuronal ROIs in vehicle (red) and CNO (green) sessions for high frequency (**f**), low frequency (**g**), 1 evidence strength (**h**), and 0 evidence strength (**i**) stimuli. Data for F-I are mean ± SEM. **j** Averaged peak neuronal ROI responses towards evidence strength +/− 1 and 0 during vehicle injected sessions ($n = 39$ neuronal ROIs; **$p = 0.0098$, two-sided Mann–Whitney U test. **k** Averaged peak neuronal ROI responses towards evidence strength +/− 1 and 0 during CNO injected sessions ($n = 36$ neuronal ROIs, n.s., $p = 0.80$, two-sided Mann–Whitney U test. **l** Change in fluorescence induced by CNO for evidence strength +/− 1 and 0 trials ($n = 16$ neuronal ROIs; * $p = 0.028$, two-sided Mann–Whitney U test. Data for J-L are mean ± SEM.

magnitudes did not vary across the different tone mixtures representing various evidence strengths (Supplementary Fig. 3j, k). All animals included in these analyzes were verified to have proper lens placement and DA2m expression in the auditory striatum (Supplementary Fig. 3l).

The finding of a negative correlation between tone-evoked striatal dopamine response and tone evidence strength (Fig. 3d, e) suggests that this dopamine activity may reflect mice' uncertainty towards the stimuli (higher dopamine response to more difficult stimulus). To investigate the behavioral relevance of evidence strength, we trained the mice to perform the same task in a reaction time–based manner with no forced wait (Supplementary Fig. 4a) while conducting DA2m dopaminergic recordings in the auditory striatum. We found that mice performed in this task tended to take longer to react to and assess high-difficulty tone presentations (Supplementary Fig. 4b, $p < 0.0001$), which is consistent with previous literature describing reaction times during decision tasks[59,60]. Similarly, in this task, we also found higher tone-evoked striatal dopamine response in trials with lower evidence strength (Supplementary Fig. 4c). There are negative correlations between striatal tonal dopamine response and evidence strength ($R^2 = −0.50$, Supplementary Fig. 4d), and between reaction time and evidence strength ($R^2 = −0.56$, Supplementary Fig. 4e). Consistently, there is a positive correlation between tonal striatal dopamine response and reaction time ($R^2 = 0.25$, Supplementary Fig. 4f). These data indicate a potential correlation between striatal tonal dopamine activity and mice's uncertainty.

We next examined whether dopamine activity is induced by a local striatal circuit or SNc cell bodies[33,61]. To monitor SNc somatic neuronal activity, we employed a genetic projection specific Ca²⁺ imaging strategy. We injected CAV-Cre into the left auditory striatum and AAV-DIO-GCaMP6f into the left SNc in a cohort of well-trained mice, followed by GRIN lens implantation above the left SNc, as illustrated in Fig. 3h, i. After 3 weeks of recovery, we recorded Ca²⁺ signals in the SNc during task performance. We recorded 59 neuronal ROIs across 20–30 sessions in six mice in which the correct viral expression pattern and lens implantation were verified during post hoc inspection (Supplementary Fig. 5a). When we aligned activity with tone onset, we detected activation in a significant fraction of the neuronal ROI (Fig. 3i), suggesting that the observed dopamine fluctuations (Fig. 3a–g) could be attributed to projection neuron activity rather than local striatal modulation[33,62]. We further analyzed changes in Ca²⁺ activity in response to evidence strengths of 0 versus 1 and found elevated Ca²⁺ signal during difficult trials (Fig. 3j, k), consistent with the striatal dopamine responses (Fig. 3d, e). Overall, in the SNc, we identified 15 of 59 neuronal ROIs that were tone-responsive in a difficulty-dependent manner, 4 of 59 neuronal ROIs that were tone-responsive independent of difficulty, and 6 of 59 neuronal ROIs that responded to varying water reward levels (Fig. 3l and Supplementary Fig. 5b–d). We should note that these reward-size responsive neurons were detected from a separated set of recordings from the same mice in which we

manipulated the water reward size (2.5 µl, 5 µl, or 10 µl) presented to mice that were well-trained in the auditory task with fixed water reward size (2.5 µl). We found these neurons do not overlap with the tone-responsive neurons in SNc.

These data indicate that during auditory discrimination behavior, tones activate auditory striatal–projecting SNc neurons, inducing dopamine release in the auditory striatum, which is positively correlated with trial difficulty.

### The D1-MSNs in the auditory striatum relay nigrostriatal dopamine modulation

Several retrograde tracing studies in other regions of the dorsal striatum suggest that nigrostriatal projections broadly target both D1 and D2 receptor–expressing medium spiny neurons (MSNs)[63], although detailed characterization remains limited. We examined the synaptic projection targets of the SNc, focusing on projections to the auditory striatum and their possible roles in auditory discrimination behavior.

We first used an anterograde, transsynaptic tracing approach to identify neuron targets of SNc projections in the auditory striatum (Fig. 4a–c). We injected an anterograde tracing virus, AAV-WGA-Cre[64], into the left SNc and a Cre-dependent fluorescent marker virus (AAV-DIO-tdTomato) into the left auditory striatum. Cre recombinase is anterogradely transported from nigral to striatal neurons to induce tdTomato expression in AAV-infected neurons. To differentiate between D1- and D2-MSNs, we performed this surgery in D2-eGFP transgenic mice, resulting in a transsynaptically labeled population expressing tdTomato and a general D2 MSN population expressing eGFP (Fig. 4a). We immunostained brain sections with an antibody against the nuclear MSN marker protein dopamine- and cAMP-regulated phosphoprotein of 32 kDa (DARPP-32; Fig. 4b[65,66]). As presented in Fig. 4b, c, although both tdTomato⁺ D1 and D2 neurons were transsynaptically labeled in the auditory striatum, substantially more D1 neurons were labeled ($p < 0.01$, two-sided unpaired t-test). Interestingly, the transsynaptically labeled neurons tended to reside in the medial subsections of the auditory striatum, a D1-MSN–rich zone, and appeared to be preferentially targeted by auditory cortical projections (Supplementary Fig. 6a[20,25,40]). The preferential transsynaptic labeling of D1-MSNs suggests that these neurons play a potential role in auditory processing and modulated by dopaminergic phasic activity.

Previously reported recordings of posterior striatal neurons, including past work from our group, were genetically anonymous (Fig. 2)[19,20]. We next determined whether tone-evoked D1-MSN and D2-MSN activities are important for auditory discrimination behavior. We employed the optogenetic approach and tone-locked silencing technique, using the bilateral expression of the neuronal silencer ArchT in auditory striatal D1-MSNs (Fig. 4d) and D2-MSNs (Fig. 4f). We found a marked impairment in task performance when D1-MSNs but not D2-MSNs were silenced (Fig. 4d, $p < 0.001$; Fig. 4f, $p > 0.05$). Given the observed behavioral role of dopamine and D1-MSN activity in the

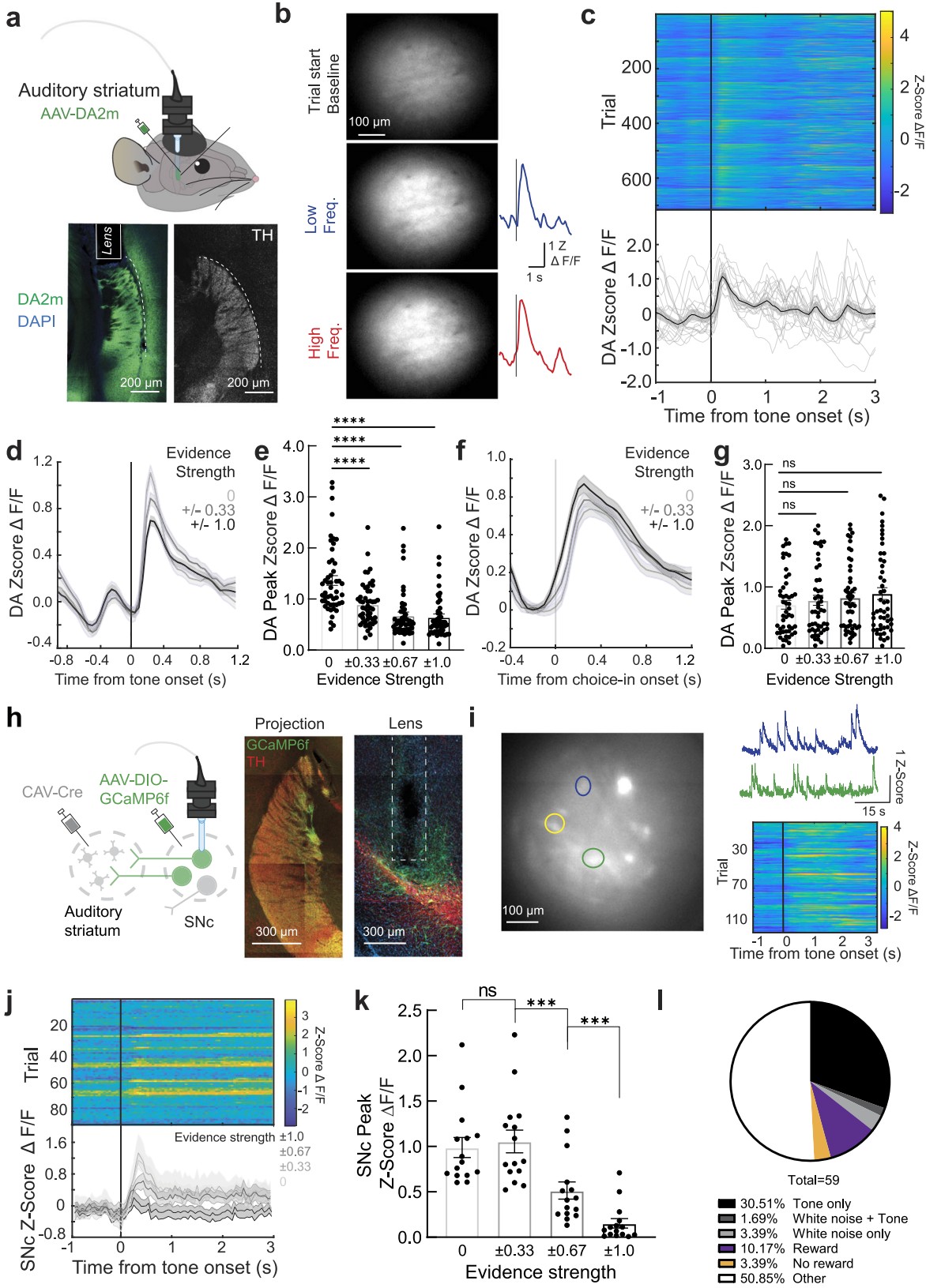

auditory striatum, we further determined whether the effects of dopamine on auditory discrimination behavior are mediated by D1 (or D2) receptors. Pre-task infusion of the D1 receptor antagonist SCH-23390 into the bilateral auditory striatum of well-trained mice (Fig. 4e) impaired task performance, suggesting that D1 receptors mediate dopamine signaling in striatal MSNs and modulate auditory

discrimination behavior. Consistent with the optogenetic manipulation effect in Fig. 4f, pre-task infusion of the D2 receptor antagonist Sulpiride into bilateral auditory striatum resulted no effect on the discrimination behavior (Fig. 4g). Mice used for these analyzes were verified to have proper ArchT expression, optic fiber placement, and cannula placement (Supplementary Fig. 6b–e). Our study indicates

**Fig. 3 | Auditory striatal dopamine fluctuates with tonal stimuli and inversely scales with evidence strength during task performance. a** Top, schematic for microendoscopic monitoring of dopamine. Bottom, lens post hoc confocal imaging of the dopamine sensor DA2m and tyrosine hydroxylase (TH) expression. **b** Representative snapshot images of whole field-of-view DA2m signal during high- and low-frequency tone presentations with corresponding session-averaged traces. **c** Representative averaged dopamine responses during tasks aligned with the tone cue presentation (n = 5 mice, 720 trials per category). **d** Averaged dopamine tonal responses as a function of evidence strength. **e** Psychometric peak dopamine response to cues as a function of evidence strength. (n = 5 mice across, with 10 sessions per animal, two-sided paired t-test ****$p < 0.0001$). Each dot presents averaged trace per session. **f** Averaged dopamine release during correct trial reward (from the time points of mice poking into the side ports) as a function of evidence strength. **g** Peak dopamine responses following reward receipt as a function of evidence strength. (n = 5 mice across 10 sessions per animal, n.s. $p > 0.05$, two-sided paired t-test). Each dot presents averaged trace per session. **h** Left, schematic for imaging auditory striatal–projecting substantia nigra pars compacta (SNc) neurons. Right, representative histology of intact dopaminergic axons in the auditory striatum and lens placement above the SNc. **i** Representative imaging field-of-view, traces, and example tone cue–responsive neuronal regions of interest (ROIs). **j** Example tone-responsive neuronal ROIs aligned with tone onset as a function of evidence strength. Top, heatmap during task session. Bottom, averaged responses as a function of evidence strength. **k** Averaged peak neuronal responses as a function of evidence strength. (Two-sided paired t-test; ***$p = 0.0005$; ***$p = 0.0006$; n.s., $p > 0.05$). Individual dots present peak values of individual SNc neurons (n = 15 neuronal ROIs). **l** Pie chart depicting the distribution of responsive neuronal ROIs. Fluctuations in dopamine sensor activity and neuronal activity are both denoted as Z-Score ΔF/F. Data in **c–g, j,** and **k** are presented as mean ± SEM.

that the inhibition of D1-MSN activity or D1 receptors, but not the D2 pathway, in the auditory striatum impairs auditory discrimination performance.

## Discussion

In this study, we characterized the dopaminergic regulation of basic functions of the auditory striatum during an auditory frequency discrimination task with embedded difficulty. Using a retrograde viral labeling strategy, we optogenetically inhibited the auditory nigrostriatal pathway in mice during stimulus presentation and found that this inhibition impaired ongoing auditory decisions. In addition, silencing the auditory nigrostriatal pathway reduced striatal sound responses during auditory decision-making. We used a microendoscopic approach to monitor dopamine dynamics and observed tone-evoked dopamine responses at both the soma and striatal terminals of SNc neurons. Dopaminergic tonal responses displayed a dynamic inverse relationship with the evidence strength of the presented stimuli, increasing in magnitude when tone clouds were more difficult to base decisions on. Finally, we determined that the D1-MSN pathway within the auditory striatum mediates the dopaminergic impacts on ongoing auditory decisions. Overall, our work reveals a critical nigrostriatal performance mechanism for supporting auditory discrimination behavior, which is further recruited by the perceptual difficulty of the task.

Experimental and theoretical studies have traditionally explored the functions of midbrain dopaminergic projections in the contexts of ventral and dorsal striatal subdivisions[26,67,68]. In this framework, the ventral striatal dopamine signals have been broadly suggested to be involved in learning processes, whereas dorsal striatal dopamine projections are suggested to regulate motor control[36,38,69,70]. In line with this idea, dopaminergic functions are likely to be even further parcellated and dependent on the fine anatomical subdivisions within the striatum. Anatomical and emerging behavioral studies have indicated the existence of specialized nigrostriatal and mesolimbic systems[22,23,35,40,41,71–74]. In this study, we focused on the auditory striatum, which is found in the tail striatum. Previous studies examining the role of dopamine in the tail striatum have demonstrated a unique response to sensory novelty[23,75], which differs from the motor functions associated with dopamine in the anterior portion of the dorsal striatum. Here, we found that familiar tones induced a phasic increase in dopamine release in the auditory striatum (Fig. 3), and the inhibition of striatal dopamine release suppressed tone-evoked striatal MSN activity (Fig. 2). Our findings not only support the notion of differential dopamine functions in subdivisions within the striatum but also provide new evidence to support a role for dopamine in sensory perception. Determining whether this role is general across sensory modalities will require future studies.

Our results showed a negative correlation between striatal dopamine release and the evidence strength of presented stimuli (Fig. 3 and Supplementary Fig. 4). Consistent with this, the nigrostriatal pathway

is important for evidence strength-dependent tonal responses in the auditory striatum (Fig. 2j–l). The stronger correlation of tonal striatal dopamine response with evidence strength than with reaction time (Supplementary Fig. 4d, f) suggests that this dopamine activity may be closer to the auditory perception than decision making. Interestingly, in previous decision-making studies investigating dopamine functions in other striatal subregions and ventral tegmental area, an opposite trend was observed (i.e., reduced dopamine neuronal or axonal firing associated with animal's low confidence towards low-evidence stimuli)[32,33,76]. One major caveat of our task is its inability to directly test the animal's perception of uncertainty or confidence[77–79]. Further exploration remains necessary to determine whether this discrepancy (opposite correlations between dopamine signal and task difficulty) is due to different animal models, striatal subregions, sensory modalities, or task designs.

Midbrain dopamine signals have been associated with reward prediction error (RPE)[80], thread prediction[23], and novel sensory stimuli detection[75]. Based on RPE theory, dopamine signals will be strongly activated by cues associated with reward, and do not respond to expected reward. In our study, we did not directly assess the striatal dopamine activity towards reward prediction error. However, we found that our recorded striatal dopamine signals are stronger to low-evidence tones (associated with a low probability of reward) and the tone-evoked dopamine activities at a given evidence strength showed no difference between correct and error trials (Supplementary Fig. 3f). The outcome-evoked dopamine activities are stronger to expected reward (correct trials) than no reward (error trials) (Fig. 3 and Supplementary Fig. 3h, i). These results suggest that the dopamine activity in the auditory striatum does not encode RPE. Consistently, SNc neurons projecting to the auditory striatum respond to tone cues in the same manner (stronger respond to tones with less evidence of strength), and the tone-responsive neurons do not respond to outcomes (reward or no reward). Most of these tone-responsive neurons do not respond to white noise which is associated with time-out punishment (Fig. 3l). Furthermore, mice used in this study were well-trained in the auditory task, so the tones are not novel cues to them. We did not observe any avoidance or defensive behaviors toward the tones from these mice. Therefore, the striatal dopamine activities analyzed in our study are less likely to be involved in novelty detection and threat prediction. Different populations of SNc neurons may mediate various striatal dopamine functions differentially. We note here, however, that our DA sensor measurements only allowed us to visualize extracellular dopamine fluctuations. It is possible that such fluctuations do not necessarily correspond to somatic downstream signaling.

Our data implicate that striatal dopamine signals correlate with the difficulty/ambiguity of the stimuli and may function as a modulator to gate striatal tone responses during discrimination behaviors. Heightened dopaminergic signaling during ambiguous cue presentation may serve to enhance striatal sensory responses to ambiguous

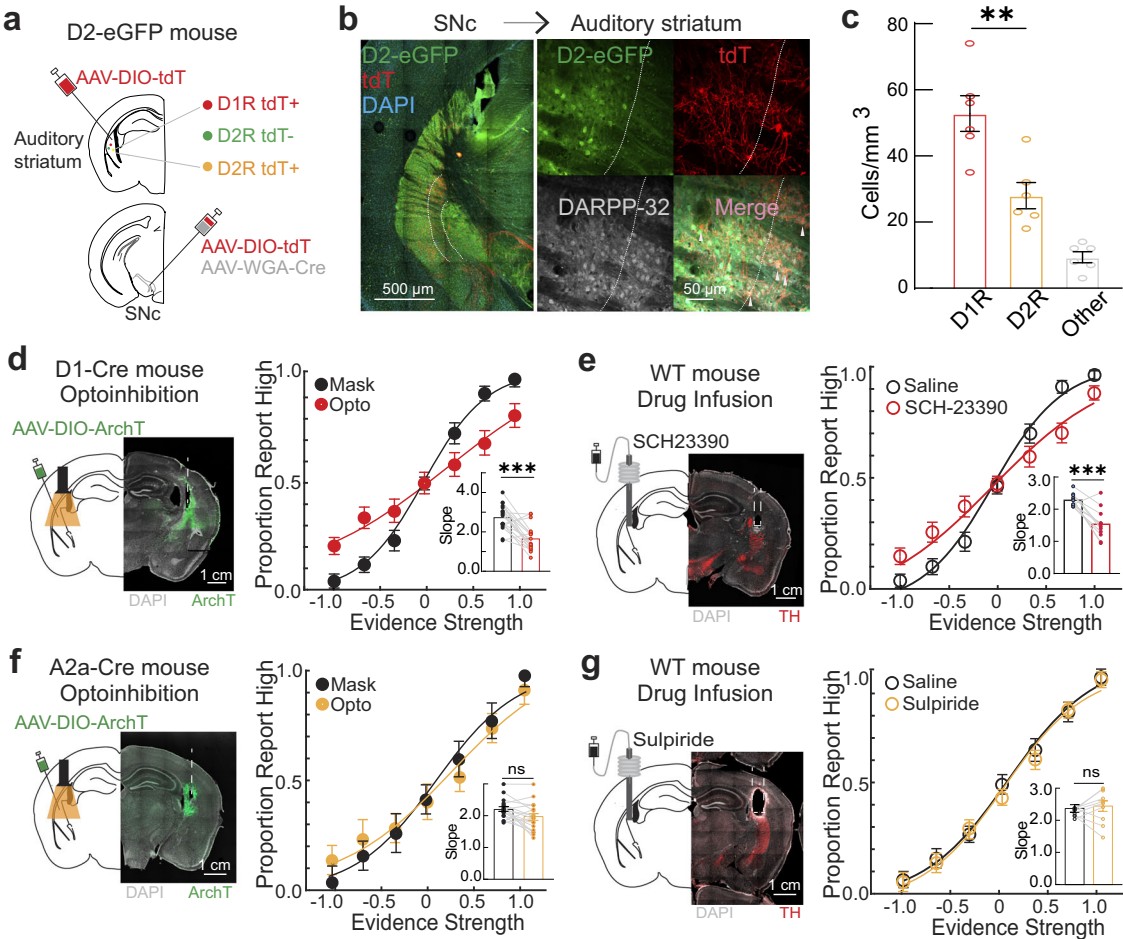

**Fig. 4 | The D1 but not D2 pathway regulates auditory discrimination.**
**a** Schematic of transsynaptic anterograde tracing from the substantia nigra pars compacta (SNc) to the auditory striatum in D2-eGFP transgenic mice. **b** Left, representative imaging of transsynaptically labeled tdTomato (tdT)$^+$ medium spiny neurons (MSNs) in the auditory striatum. Right, imaging of tdT + , DARPP-32 + (MSN marker), and D2-eGFP+ neurons. White traces denote the D1R-rich zone. **c** Quantification labeled neuron distribution ($n = 18$ sections across 6 mice; **$p = 0.0040$, two-sided unpaired $t$-test). Each dot represents a density per mouse. **d** Left, schematic for optogenetic silencing of D1-MSNs and representative histology. Right, psychometric performance during auditory stimulus–based silencing comparing optogenetic light vs. masking light trials. ($n = 4$ mice across 18 sessions). Insert: comparison of corresponding regression slopes for individual sessions. Bars represent the mean and dots represent individual sessions (***$p = 0.0002$, two-sided Wilcoxon rank-sum test). **e** Left, schematic for bilateral drug microinfusion of D1R antagonist and representative histology. Right, psychometric performance

comparing saline vs. D1R antagonist sessions. ($n = 4$ mice across 12 sessions). Insert: comparison of corresponding regression slopes for individual sessions. Bars represent the mean and dots represent the individual sessions (***$p = 0.0005$, two-sided Wilcoxon rank-sum test). **f** Left, schematic for bilateral optogenetic inhibition of the D2-MSNs. Right, psychometric performance during auditory stimulus–based silencing comparing optogenetic light vs. masking light trials. ($n = 4$ mice across 18 sessions). Insert: comparison of corresponding regression slopes for individual sessions. Bars represent the mean and dots represent the individual sessions (n.s., $p = 0.1453$, two-sided Wilcoxon rank-sum test). **g** Left, schematic for bilateral drug microinfusion of D2R antagonist and representative histology. Right, psychometric performance comparing saline vs. D2R antagonist sessions. ($n = 4$ mice across 10 sessions). Insert: comparison of corresponding regression slopes for individual sessions. Bars represent the mean and dots represent the individual sessions (n.s., $p = 0.3150$, two-sided Wilcoxon rank-sum test). Data in **c**–**g** are presented as mean ± SEM.

stimuli. Dopaminergic neurons that project to the tail striatum may receive signals associated with acoustic information and its subjective difficulty through differential inputs such as the orbitofrontal cortex and the subthalamic nucleus, where have been implicated in perceptual difficulty in decision-making contexts[81–83]. Explorations of anterograde tracing from the primary auditory cortex performed by our group and others indicate that the auditory cortex projects directly to the lateral portion of the substantia nigra (Supplementary Fig. 6f and Allen Brain Atlas). Stimulus ambiguity may be encoded directly in the auditory cortical network, and a large body of work suggests that the auditory system is differentially activated when animals engage with naturalistic or more complex acoustic information[1,84–86]. Future work will be required to dissect the inputs that control auditory striatal dopamine computations, as has been performed for other circuits[87].

In this study, we probed the functions of the D1-MSN and D2-MSN populations of the auditory striatum in auditory discrimination. We

found that although both D1-MSNs and D2-MSNs receive synaptic inputs from SNc, only inhibition of D1-MSNs and D1 receptors affect mice' discrimination performance in this auditory task (Fig. 4). This is consistent with prior literature demonstrating a function of the D1-MSN pathway in auditory choice behavior[19], whereas the D2-MSN has been indicated a role for different types of learning and value updating[66,88–90]. Prior studies have indicated that the high-affinity D2 receptors have an optimized role in detecting DA 'dips' or omissions of DA, whereas the low-affinity D1 receptors are optimized for detecting phasic increases in DA[91,92]. While our current study indicates that the D1 receptor pathway in the auditory striatum primarily facilitates dopamine's role in phasic auditory discrimination, future studies will be required to uncover the potential sensory function of the D2 receptor pathway in this area.

Overall, our study details a neuromodulatory mechanism for ongoing auditory cognition, which advances the understanding of how midbrain dopaminergic signaling modulates sensory output.

## Methods

### Animals

All animal procedures were approved by the Stony Brook University Animal Care and Use Committee and were performed in accordance with the National Institutes of Health standards. C57BL/6 J (The Jackson Laboratory), DAT-IRES-Cre (The Jackson Laboratory, 006660), Ai14 (The Jackson Laboratory, 007914), D1R-Cre (The Jackson Laboratory, 37156), and D2-eGFP/rpl10a (The Jackson Laboratory, 030255), A2a-Cre (MMRRC, 036158 UCD) mice were used for this study. Both male and female 2–4-month-old mice were used for this study. Mice were housed with free access to food but were water-restricted after the start of behavioral training. Animals were housed in a 12-hour light/dark cycle, and all behavioral experiments were conducted during the animal's dark cycle. On training days, water was made available in response to task performance. Mice retrieved 2.5 μl water for each correct trial and were ensured at least 1 mL water each day. On non-training days, water bottles were provided to the mice for at least 1 hour.

### Stereotaxic procedures and viral injections

Mice (7–9 weeks old) were anesthetized with 4% isoflurane and placed into a stereotaxic apparatus with continuous 1% isoflurane delivery. According to previously described procedures[18,20], viral injections and the implantation of optic fibers, lenses, and cannulae were performed at the following stereotaxic coordinates anteroposterior (AP) and mediolateral (ML) to bregma and dorsoventral (DV) relative to the cortical dural surface, as measured with the descending object's tip: auditory striatum: −1.7 mm AP, ± 3.30 mm ML, − 2.5 mm DV; substantia nigra pars compacta: −3.10 mm AP, ± 1.50 mm ML, and 4.2 mm DV; auditory cortex: −2.8 mm AP, ± 4.15 mm ML, − 0.9 mm DV; and dorsolateral striatum, +0.5 mm AP, ± 2.10 mm ML, 2.4 mm DV.

First, a 0.5–1.0 mm diameter craniotomy was performed over the desired coordinates using a dental drill instrument. A custom glass micropipette with a tip diameter of 10–15 μm was loaded with 1 μl viral solution and slowly inserted into the brain until reaching 0.2 mm ventral to the target location. The micropipettes were slowly retracted dorsally to the actual target coordinates prior to viral injection. Viral solutions were delivered through the glass pipette connected to a Picospritzer II microinjection system (Parker Hannifin Corporation) at a rate of 100 nl/min. For striatal injections, approximately 300–500 nl of viruses were injected. For SNc injections, approximately 500–800 nl of viruses were injected. After the viral delivery, the needles were maintained in position for 10 minutes, followed by a slow retraction at a rate of -10 μm/s. The following viruses were used: retrograde axonal tracing from the striatum or retrograde optogenetic/imaging experiments: CAV2-Cre and CAV2-GFP (PVM, France); optogenetic silencing: AAV2/9-CAG-DIO-ArchT-GFP (UNC vector core) and AAV2/9-CAG-EGFP (Addgene, #37825) as experimental and control viruses, respectively; ex vivo optogenetic activation: AAV2/5-CAG-hChR2(H134R)-mCherry (Addgene, #100054); dopamine sensor imaging: AAV2/9-CAG-DA2m (WZ Bioscience, US; Yulong Li lab) and AAV2/9-CAG-EGFP (Addgene, #37825) as experimental and control viruses, respectively; calcium sensor microendoscopic imaging: AAV2/9-CAG-GCaMP6f (Addgene, # 100836) and AAV2/1-hSyn-DIO-GCaMP6f (Addgene, #100837); designer receptor exclusively activated by designer drugs (DREADD)-based chemogenetic silencing: AAV2/8-hSyn-DIO-M4Di-mCherry (Addgene, #44362) and AAV2/8-hSyn-DIO-mCherry (Addgene, # 50459) as experimental and control viruses, respectively; anterograde tracing from the SNc: AAV2/9-WGA-Cre (custom-made) and AAV2/8-DIO-tdTomato (Addgene, #28306). All subsequent optic fiber and lens implantations were performed simultaneously with viral injections. Optic fibers and cannulae were implanted bilaterally, whereas lens implantations occurred unilaterally on the left hemisphere.

### Behavioral task

Mice were water-restricted and trained to perform an auditory, two-alternative, forced-choice discrimination task to retrieve water, as described previously[20]. Briefly, behavior was centrally mediated and continuously monitored using Bpod, an open-source MATLAB-based behavioral control system (Mathworks, Natick MA; Sanworks, Rochester NY). Freely moving mice self-initiate a trial by poking their nose into the center port of a three-port chamber located within a sound-attenuated behavioral rig. Center poking triggered a 1.5 s phase consisting of a 500 ms pre-cue delay, a 500 ms cloud-of-tones sound cue, and a 500 ms post-cue delay. Withdrawal from the center port during this time period resulted in an early withdrawal sound (White noise) punishment and a timeout of 5 s. In the reaction time version of this task, mice were subjected to a 500 ms pre-cue period but were free to withdraw from the center port at any time with no imposed post-cue delay. The cloud-of-tones cue consisted of a 30-ms stream of overlapping pure tones presented at 100 Hz. The overlapping tones were drawn from 18 possible tone frequencies in the 5–40 kHz range. Low-frequency tones: 5–10 kHz; high-frequency tones: 20–40 kHz. Evidence strength = (number of high tones − number of low tones)/(number of high tones + number of low tones). For an evidence strength of 0, the left or right port had equal probabilities of being the correct choice. For these datasets, mice were trained to arbitrarily associate a low-frequency target tone with the left port and a high-frequency target tone with the right port. Correct responses were rewarded with water (2.5 μl), and incorrect responses were punished with a 4 s timeout. The sound intensity for all tones was calibrated to 60 dB (Bruel and Kjael).

In a subset of recordings, we probed how SNc neuronal activity changes as a function of reward sizes. In these recordings, we manipulated the water reward sizes (2.5 μl, 5 μl, or 10 μl) presented to mice that were well-trained in the auditory task with fixed water reward size (2.5 μl).

### Behavioral data analysis

Quantification of behavioral and psychometric performance was performed as previously described[16,20]. In brief, the evidence strength, termed r, was determined as the difference between the rate of low-frequency (5–10 kHz) and high-frequency (20–40 kHz) tones presented in each stimulus cue. Tones were drawn from the target range with a probability of $1 + 2 \times r/100/3$. To generate psychometric curves, we employed a logistic regression model using the MATLAB-based function *FitPsycheCurveLogit*[16,20]: $\log (p / (1-p)) = \beta 0 + r \times \beta 1$, where p is the fraction of choices associated with the right, high-frequency target port. The terms β0 and β1 refer to the bias and slope of the psychometric curve, respectively. Choice movement time was calculated as the time between center port withdrawal and choice-port entry, based on timestamps. The early withdrawal rate was calculated based on the portion of trials during which the animals withdrew from the center tone presentation port at any point during the 1.5-s forced wait (0.5 s for pre-tone delay, 0.5 s for tone presentation, and 0.5 s for post-tone delay). For the reaction time version of the task, the reaction time was calculated as the time between the tone onset and the time to withdrawal from the center port.

### Optogenetic experiments

Experiments were performed as previously described[20]. Briefly, cannulated optic fibers (Thorlabs, US) were hand-made and polished to produce a 10 mW, 530 nm laser output using a solid-state laser (Shanghai Dream Lasers, Shanghai, China). Similar surgical procedures were performed for auditory striatal–projecting SNc cell body inhibition, SNc terminal inhibition, and D1R MSN optical inhibition. Well-trained animals were subjected to simultaneous bilateral DIO-ArchT (with bilateral CAV-Cre infusion into the auditory striatum for cell body inhibition) or control DIO-eGFP viral injection and optic fiber implantation 400 μm above the SNc or the auditory striatum. Optic fibers

were then slowly lowered at a rate of approximately 10 μm/sec. Implantations were secured using UV-cured white dental cement (AC Flow-It), cyanoacrylate, and dental cement (Stoelting). Mice were allowed to recover for 1 week and returned to training for at least 3 weeks to allow for viral expression. For training and experimental days, an FC/PC patch cord using a FiberPort Collimator (Thor Labs) was connected to the implanted optic cannulae. Light onset was controlled by Bpod and PulsePal (Sanworks), which were connected to the laser output system. Continuous light pulses were delivered starting 500 ms before the sound cue onset and lasted for 1000 ms to cover the entire period of the sound cue presentation (500 ms total). A masking light placed above the central port delivered a light pulse at 530 nm in a similar manner. Manipulation trials (5–10% of trials) were randomly interleaved with control masking trials.

### In vivo imaging experiments

Microendoscopic procedures were performed as previously described[8,53]. Briefly, GRIN lenses 7.3 mm (Inscopix Inc., Palo Alto CA, 1050-002179) or 6.1 mm in length (Inscopix Inc., 1050-002182) were used for both auditory striatal and SNc imaging. Animals underwent simultaneous viral injections and lens implantation surgeries, with lenses placed 200 μm above the viral injection coordinates on the left auditory striatum or the left SNc. A 3D-printed metal lens holder with a stereotaxic attachment was used to assist with lens implantation. Lens implants were subsequently secured using UV-cured, white dental cement (AC Flow-It), cyanoacrylate, and dental cement (Stoelting). Prior to lens implantation, 300–500 μm cortical tissue was carefully aspirated, preventing excessive bleeding. The lens was then slowly lowered at a rate of approximately 10 μm/sec. For SNc surgeries, the lens was lowered by 1 mm, retracted ventrally 100 μm, advanced back to the original coordinates, and allowed to settle for 2 minutes. After surgeries, mice were allowed to recover for at least 3 weeks prior to baseplate implantation. Baseplate implantation timing was guided by the identification of clear neuronal ROIs. After baseplate implantation, animals were allowed 3 days to recover before habituation to a mounted microendoscopic camera. During habituation to the mounted camera, animals were allowed to perform the auditory task until they achieved pre-surgical baseline performance (for at least one session) prior to imaging. Imaging data were acquired using the nVista2.0 hardware system and software (Inscopix Inc.). Behavior was controlled using a Bpod-based behavioral controller (Sanworks), which was programmed to send behavioral timestamps using BNC signals to the nVista system. Imaging sessions were limited to 12 minutes per session to avoid photobleaching. For all imaging sessions, videos were acquired at a 20-Hz frame rate. Depending on the basal fluorescence level, the power intensity of the camera was set at 10–40% LED power, with a digital gain of 2–3.

### Simultaneous chemogenetic and microendoscopic imaging experiments

Similar methodologies were applied when performing simultaneous DREADD-based silencing and microendoscopic imaging for both dopamine sensor (DA2m) and neuronal calcium (GCaMP6f) imaging datasets. The viral infusion of DA2m or GCaMP6f constructs, Cre-dependent hM4Di constructs, and lens implantation in the left auditory striatum occurred simultaneously in DAT-IRES-Cre mice. Mice were allowed to recover for at least 3 weeks before baseplate implantation. After recovery, mice were mounted with the endoscopic camera and allowed to habituate to the camera presence during task performance. After returning to baseline performance, mice were habituated to i.p. injections of 300 μl water (vehicle) 30 minutes prior to starting each day's task session. Mice were subjected to vehicle injections until baseline performance was achieved. Subsequently, the mice underwent cycles (one session per day) of vehicle or CNO (Enzo;

i.p. 2.5 mg/kg, 300 μl with water diluent) injections 30 minutes prior to in vivo imaging sessions.

### In vivo imaging data analysis

All imaging data sets were first processed spatially, down-sampled, and rigid-based motion-corrected (Mosaic, Inscopic). For the DA sensor and GFP control data, an overall maximal ROI was drawn using the Mosaic software to determine the continuous ΔF/F signal across each session, using the mean fluorescence across each individual session as the F0 component. For calcium imaging data, neuronal ROI activity was extracted using the MATLAB-based CNMF-E algorithm[93]. Specifically, we used a probabilistic cellular registration algorithm which estimates the probability of correct registration of a spatial footprint across multiple imaging sessions. A pair of cells is considered to have the same identity if the calculated probability is $p > 0.5$ with a centroid distance of 5 um. Spatial components using this method were subsequently inspected for cellular shape and similarity of calcium transient dynamics across behavioral sessions. Manual inspection was used to determine and verify whether non-overlapping neuronal ROIs were extracted. The extracted $C\_raw$ component was used to determine the ΔF/F metric and was used for subsequent analyzes. The deconvolved calcium signal, $S$, was used to register calcium event rates. Cellular registration was performed by applying a previously developed cell tracking algorithm to the spatial component of the CNMF-E datasets[56]. Bpod-based timestamps were used to align the calcium transients (extracted temporal components) with various behavioral events, such as the center poke, tone onset, or reward onset. All datasets were analyzed using MATLAB, with further statistical analyzes performed using GraphPad Prism 8 (Graphpad Software Inc., San Diego, CA).

### In vivo validation of DA sensor imaging

It has been previously validated that in vivo i.p. injection of eticlopride can blunt nigrostriatal-induced DA fluctuations[57]. Here, we used a similar method and performed i.p. injections of either saline or eticlopride dissolved in saline (1.0 mg/kg, Cayman Chemicals, Ann Arbor MI) in well-trained mice 15 minutes prior to placing mice in behavior rigs and recording setups. Recordings were performed for a continuous 8-minute period and tone-evoked responses were analyzed in the same manner as with all other in-task recordings.

### Ex vivo validation of DA sensor imaging & SNc neuronal silencing

To validate nigral control of DA levels and thus DA2m-based fluorescence, we employed an ex vivo approach described previously[94]. We infused AAV9-CAG-DA2m in the auditory striatum and AAV5-CAG-ChR2 in the SNc of wild-type mice. 3-5-week after viral infusion, mice were anesthetized with a ketamine/xylazine mixture (100 mg/kg and 7 mg/kg, respectively) and underwent transcardial perfusion with ice-cold artificial cerebral spinal fluid (ACSF) consisting of 124 mM NaCl, 3 mM KCl, 1 mM CaCl2, 26 mM NaHCO3, 1 mM NaH2PO4, 14 mM glucose, and 1.5 mM MgCl2. 275 μm para-sagittal slices were obtained using a vibratome (VT1000s; Leica Microsystems, Buffalo Grove, IL). The auditory striatum's location was determined based on the sagittal anatomic coordinates and expression of DA2m. Slices were subsequently placed in the recording chamber of a two-photon Ultima Laser Scanning Microscope System (Bruker Nano, Inc., Middleton, WI) and bathed with ACSF solution (containing 2 mM CaCl2 and 1 mM MgCl2) at room temperature for recording. To activate ChR2-infected SNc terminals in the auditory striatum, five 1 ms duration optical stimulations were delivered at 30 Hz using a single photon 473 nm laser tuned to a power of 3 mW (Coherent OBIS FP 473LX, Coherent, Inc., Santa Clara, CA). Spiral line scans (920 nm, 21.2 ms with 0.0776 μm 2 pixels and 10 μs dwell) were performed to measure DA2m fluorescent changes. A high-speed shutter was used to block cross-signaling between optogenetic stimulation and DA2m fluorescence measurements. Slices

were either bathed in ACSF or ACSF with dissolved Haloperidol (10 μM; Sigma Aldrich),

For validation of SNc optogenetic neuronal silencing, we infused AAV-CAG-DIO-ArchT-GFP in the SNc of DAT-Cre mice. Acute brain slices were prepared as described above. The SNc's location was determined via expression of ArchT-GFP. A sustained 5 s optical stimulation was used to silence ArchT-infected SNc neurons. Fluorescently labeled soma of putative DAergic neurons in the SNc were targeted for cell-attached voltage clamp recording of pace-making activity followed by the sustained optical stimulation. Current clamp recordings were also obtained after patch rupture to confirm the strong hyperpolarizing effect of ArchT.

### Cannulae D1R and D2R inhibition experiments

Guide cannulae were implanted bilaterally 2.0 mm ventral to the cortical surface at the AP and ML coordinates previously described for the auditory striatum (26-gauge, 4.0 mm, Plastics One)[20]. Briefly, the cannulae were slowly lowered at a rate of approximately 10 μm/s and were affixed using white UV-cured dental cement and acrylic dental cement (Lang Dental Manufacturing, IL). Dummy cannulae were placed within the guide cannulae to protect the underlying brain tissue. Mice were allowed to recover for 1 week, after which the animals were returned to training. After task performance became stable and returned to pre-surgery levels, mice underwent vehicle and drug performance cycles, in which mice were infused with 300 nl of saline or 1 mg/ml D1R antagonist SCH-23390 diluted in saline (Cayman Chemicals, Ann Arbor, MI) 30 minutes prior to task sessions.

Similar to the D1R antagonism experiments, we performed D2R antagonist cannula infusions in well-trained mice. For these experiments, we used a D2R-specific antagonist, Sulpiride (Cayman Chemicals, Ann Arbor, MI), previously tested in microinfusion experiments on behaving mice[95,96]. Briefly, either 300 nl of saline or 300 μM Sulpiride (dissolved in saline) was infused bilaterally to the auditory striatum. Infusions were performed 30 minutes prior to the behavioral task.

### Histology and Immunostaining

Mice were anesthetized using 4% isoflurane through chamber delivery, followed by urethane injection (i.p., 250 μl/g). Mice were subsequently transcardially perfused with ice-cold physiological phosphate-buffered saline (PBS) and 4% paraformaldehyde (PFA). Brains were extracted and fixed in 4% PFA for 16–24 h. Brains were then washed with PBS three times prior to coronal sectioning (50–70 μm) using a vibratome (Leica Microsystems). Free-floating immunofluorescence staining was initiated by washing sections in PBS three times and blocking with 1% donkey serum in 0.25% PBST (PBS + 0.25% Triton-X) for 1 h at room temperature. The sections were then incubated with the following primary antibodies in 0.25% PBST with 1% donkey serum: rabbit anti-TH (1:1000; Abcam), mouse anti-TH (1:1000, Millipore), mouse anti-DARPP-32 (1:1000, Santa Cruz Biotechnology), goat anti-GFP (1:1000, Rockland), and rabbit anti-red fluorescent protein (RFP; 1:1000, Rockland). Primary incubation occurred at 4 °C overnight. Sections were then washed three times in 0.25% PBST and subjected to secondary antibody incubation in 0.25% PBST: donkey anti-rabbit 647 (1:1000, Jackson ImmunoResearch), donkey anti-rabbit 594 (1:1000, Jackson ImmunoResearch), donkey anti-mouse 594 (1:1000, Jackson ImmunoResearch), donkey anti-mouse 647 (1:1000, Jackson ImmunoResearch), and donkey anti-goat 488 (1:1000, ThermoFisher). Sections were mounted using Fluormount-G™ (ThermoFisher) and imaged using a Zeiss confocal microscope (Zeiss LSM 800). For anterograde and retrograde tracing experiments, RFP/TH/GFP/DARPP-32⁺ neurons were manually counted based on neuronal morphology and the presence of 4′,6-diamidino-2-phenylindole (DAPI) nuclear signaling using ImageJ software.

### Quantification and statistical analysis

All data were processed and analyzed in MATLAB and GraphPad Prism 8 (Graphpad Software Inc.). Where appropriate, datasets were tested for normality using the Kolmogorov–Smirnov test. Paired or unpaired t-tests were used to analyze datasets assumed to conform to normal distributions. One-way ANOVA was used to assess differences in reaction time in the reaction time version of the task. The Wilcoxon rank-sum or Mann–Whitney U test was used to analyze datasets that did not meet the normality assumption. Unless otherwise noted, all data are reported as the mean ± standard error of the mean (SEM). To determine whether a particular fluorescence signal was significantly responsive toward a behavioral variable (e.g., tone cue–responsive), the mean value of the continuous ΔF/F signal over a 0.5-s period after the behavioral timestamp was tested for significant difference from the mean value 0.3 s before the time-stamp onset (Wilcoxon rank-sum test at $p < 0.05$). Although our sample sizes were comparable to previous literature using similar experimental paradigms, statistical methods were not utilized to predetermine appropriate sample sizes[3,20,38]. For behavioral and imaging analyzes, experiments were not blinded for conditions, as all parameters were objectively measured using imaging or behavioral software. All animals included in the study that required implantations were ensured to have proper and reproducible implant placements (implantation sites indicated in respective supplementary figure cartoons, e.g., Supplementary Fig. 1A).

### Reporting summary

Further information on research design is available in the Nature Research Reporting Summary linked to this article.

## Data availability

All data in the manuscript is either in the source data file or presented within the figures of the manuscript. All the data that support the findings of this study are available upon reasonable request. Source data are provided with this paper.

## Code availability

The custom codes used for data analysis in this study are attached in the supplementary file.

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

## Acknowledgements

We would like to thank Drs. Anissa Abi-Dargham, Robert Froemke, Adam Kepecs, Yang Yang, and Ge and Xiong laboratory members for their valuable comments on the manuscript. This work was supported by the National Institutes of Health [DC016746 and DC017470 to Q.X.; F30DCDC018214 to A.P.C.; NS089770, AG046875, and NS104868 to S.G; NS104089 to J.L.P], and Stony Brook University startup funding (to Q.X.).

## Author contributions

A.P.C., S.G., and Q.X. designed the experiments. A.P.C. performed most of the experiments and data analysis. Exceptions to this include L.C. and J.M.M. performance of dopamine sensor validation in vivo and ex vivo experiments. L.C., K.S., and E.C. partially performed with behavioral imaging recordings and animal training. J.L.P. designed the ex vivo two-photon imaging experiments, which were performed by J.M.M. A.P.C., S.G., and Q.X. wrote the manuscript.

## Competing interests

The authors declare no competing interests.
