## [Peer Review File · Nature Communications]

Nigrostriatal dopamine pathway regulates auditory discrimination behaviorREVIEWER COMMENTS

Reviewer #1 (Remarks to the Author):

Chen et al. investigated the tail portion of the striatum (TS) in an auditory discrimination task using a two-alternative forced choice paradigm. Optogenetic inhibition of either the cell bodies or the terminals of the midbrain dopaminergic projection to the tail of striatum impaired task performance. Next, the Authors found by in vivo Ca imaging that a portion (about 15%) of TS neurons respond to sound. Surprisingly, sound responses were greater when the tones did not convey sensory evidence as compared to easy trials. Chemogenetic inhibition of dopaminergic neurons abolished these tone responses. In agreement, DA release in the TS and dopaminergic cell bodies in the SNc showed a similar activation pattern. Finally, D1 MSNs in TS were also shown to be tone-responsive and necessary for task performance.

The TS represents a new hot topic and the Authors approached it from an unusual angle of perceptual decision making, making this an exciting and important study. The questions were addressed by multiple and adequate methods with many important control experiments; thus, the experimental design was thorough and appropriate. I have some reservations about data analysis and presentation: I believe the data could be better analyzed to embed the results in the context of the newly emerging TS field as well as the long-established dopamine field, thus engaging a broad audience. I think my points can be addressed by more in-depth analysis of the data already collected, after which this paper could represent an important advance in our understanding of the dopaminergic control of striatal circuits.

Major points

1. The Authors characterize tone responsiveness of TS neurons. What I mostly miss from their analysis is a better embedding of their results into the vast literature of DA control over striatal functions, as well as into the newly emerging literature of TS.

1a. Tone-responsive neurons constitute only 15% of TS. Do the other TS cells display different task-related activity, or no task-related activity at all? Is this proportion considered high compared to other parts of the striatum?

1b. Do TS neurons respond to reward? Does a part of TS neurons show RPE activity? The major finding of the negative correlation between DA input and evidence strength contradicts the RPE theory, which would predict stronger DA responses for cue stimuli of stronger reward-predictive strength. Some analysis is presented in Figure S3H, but the grand averages do not provide a clear picture. Can the larger DA responses at zero evidence be related to a larger probability of error and consequential time out, as some form of 'threat prediction' (see relevant papers of the Uchida lab)? In Fig. 3L, how were the 'reward prediction error' neurons defined? Tone responsive is supposed to be 32% according to the label, but that is at odds with the pie chart, where it looks very close to 25%. What does 'responded to varying water reward levels' mean? I was surprised to read this, as I could not find previous mentions of different reward sizes. This seems important in a task where neurons potentially related to RPE are probed, so this should be clarified. Were 'tone responsive' and RPE neurons non-overlapping? This is unclear, as I would expect RPE neurons to be tone-responsive as well in this task. Does this mean that tone responsive neurons are not feedback responsive or just not correlated with the reward size? What are the neurons in Fig. 3K? All recorded, 'tone responsive', or 'tone-responsive in a difficulty dependent manner'? How do reward responses look as a function of evidence strength in the different subpopulations? How do different cells respond in error trials? Does this depend on evidence strength? This could be interesting for both SNc responses and DA sensor data.

1c. Do TS and SNc DA neurons respond to the sound punishment for early withdrawal? How do responding neurons overlap with those categorized as tone responsive? Is there any activity related to

the time-out punishments? The tones used here were 60 dB, which are borderline aversive. Do the Authors have evidence that they were not perceived as aversive, and the tone responses were not related to sound intensity? Especially given that tone-evoked dopamine sensor responses in the passive task seemed to be around the same magnitude as in the active version of the task, which is unlike most cue-evoked DA responses that emerge through the training. These questions seem important in light of data from the Uchida lab, which characterizes the system at study as 'threat prediction error' representation. A more thorough discussion of this literature and its relation to the present findings could help a lot in interpreting the results.

2. The Authors showed D1 MSNs may be necessary. It should be emphasized that this does not rule out the potential role of D2 MSNs too strongly. In this regard, in Fig.2C, were the counts normalized to relative density of D1 and D2 MSNs in TS? That is, is this truly preferential targeting? Relatedly, if I'm not mistaken, since all TS had 15% tone responsive population, and so had D1 MSNs, then necessarily D2 MSNs have the same 15% tone responsive subpopulation.

3. The Authors make a point in the discussion that their results contradict a role of the TS DA input in learning. However, mice were overtrained and learning was not directly tested. Therefore, I think this claim should be removed, since in the present form, it is not implicated by the data.

4. The Authors rightly realize that their results may be linked to confidence representations. They conduct an additional experiment; however, this is relegated to the Discussion section and remains quite obscure. First, the corresponding paragraph in page 17 is unclear. What question was addressed and how, what 'discrepancy' remained unresolved (line 7-8)? I think this experiment could be better presented in the Results, with some additional analyses that could shed some more light on the nature of the neuronal representations. Specifically, evidence strength, reaction time and peak DA responses were all correlated. Which of these three associations were stronger? Can computing partial correlations or mutual information help understand the relationship of these variables?

Minor points:

Chemogenetic inhibition in Fig.2 seems to largely abolish tone responses. Therefore, it seems rather trivial that differences related to task difficulty (evidence strength) also disappear. Maybe this part could be rephrased to make this clearer.

It is hard to see the point of Figure S3F and S3G for those trials in which the stimuli were of zero-evidence, as both ports had equal chance of being the correct port. On the other hand, I would be curious whether this figure looks the same for the other trials.

p. 4. line 11 'dorsal that' should probably read 'dorsal striatum that'; p 13 line 16 'zone' not 'zones'

Fig. S3L, S5B-D are barely visible.

Fig. 3F - What is choice-in onset? Please explain or find a better label.

Fig. 3I - Yellow trace is not visible.

Fig. S4 shows an example neuron. Adding averages across cells could make this figure more convincing.

Fig. S4B: x axis 3s tick label is misplaced.

Fig. S5 title should be rephrased

Discussion: 'Our results showed a positive correlation between striatal dopamine release and the evidence strength of presented stimuli (Fig. 3).' Did you mean negative correlation?

I believe that openness and transparency can increase the fairness of peer review. Therefore, I decided to sign my reviews.

Balazs Hangya

Reviewer #2 (Remarks to the Author):

Chen et al. have investigated a nigrostriatal dopamine pathway and its role in regulating auditory discriminatory behaviors. The authors demonstrated that the optogenetic inhibition of nigrostriatal dopaminergic projections specifically impairs the mouse choice performance but not a movement in an auditory frequency discrimination task. The authors also test the hypothesis that increasing the task difficulty influences the dopamine dynamics in the auditory striatum. To test their hypothesis, they use a combination of the microendoscopic approach and the dopamine fluorescence sensor (the G-protein-coupled receptor activation-based series DA2m).

The authors indicated that they delivered optical stimulation during both pre-tone and tone phases to achieve sufficient silencing during tone presentation.

Can the authors provide further evidence that the optogenetic suppression produced a silencing of the neurons in the SNc?

The authors should provide more results about when the auditory nigrostriatal neurons are active during the behavioral task (for example, extracellular recording). This will give the base to manipulate the activity (activation/silencing) of the SNc neurons during the behavioral task.

The authors performed an experiment to verify that this phasic response represents physiological changes in dopamine. They injected the D2R-specific antagonist eticlopride and observed blunted tone-evoked DA2m responses in mice during cued tasks. In addition, the external application of another D2R-specific antagonist, haloperidol, diminished the increase in DA2m fluorescence evoked by SNc axon terminal activation using channelrhodopsin-2 in acute brain slices (Supplementary Fig. 3D). Can the authors provide more evidence in the results section of how this experiment was performed?

Chen et al. study indicated that D1-MSNs in the auditory striatum respond to tone stimulation during the auditory discrimination task, and the inhibition of D1-MSN activity or D1 receptors in the auditory striatum impairs auditory discrimination performance.

Their experiment used an anterograde transsynaptic tracing approach to identify neuron targets of SNc projections in the auditory striatum. This experiment indicated that both tdTomato+ D1 and D2 neurons were transsynaptically labeled in the auditory striatum.

For this reason, it is crucial for this study to demonstrate the role of D2-MSN neurons such as examine whether tone-evoked responses occur in the D2-MSNs during the auditory task and determine whether tone-evoked D2-MSN activity is essential for auditory discrimination behavior.

Reviewer #3 (Remarks to the Author):

In the manuscript entitled "Nigrostriatal dopamine pathway regulates auditory discrimination behaviour" by Chen et al., the authors use a forced choice tone discrimination task in mice to investigate the role of dopamine projections to the tail of the striatum, referred to "auditory striatum" by the authors. They find that optogenetic inhibition of the tail of the striatum during the choice period impairs discrimination of tones. They then demonstrate that chemogenetic inhibition of striatonigral dopamine projections reduces tone responses in the tail of the striatum. Direct imaging of DA activity at the soma and release at the terminals in the tail of the striatum, revealed DA responses during tone presentation. Finally, they authors show that behavioral performance involves D1- but not D2-receptor neurons.

The manuscript includes an impressive series of experiments, using state-of-the-art tools to address an interesting question. Building on previous findings that D1-receptor neurons in the tail of the striatum are involved in performance in the perceptual choice task, the new experiments uncover a function of dopamine in mediating striatal tone-related neural activity and behavioural performance.

Major comments:

1) The author's interpretation of the observed neural responses raises several questions for me. For the results shown in Figure 2, the authors seem to favour the interpretation that neural responses to tones in the tail of the striatum are related to "general perception" (p.17 l.9). I find this conclusion a bit difficult to follow. While I don't question that neurons are activated by sounds in the task, it is not clear from the analysis done what these neurons actually respond to. What is the response selectivity of neurons to different sounds (e.g. Guo et al. 2018) ? Are there auditory responses in the tail of the striatum of untrained animals?

Similarly, the interpretation of DA responses shown in Figure 3 is not entirely clear. What are DA neurons encoding ? DA response to sensory stimuli that predict a possible reward are generally consistent with RPE. Maybe the correlation of DA and evidence could be related to the higher stimulus complexity of the low-evidence sound cloud or to (cognitive) effort ? Also, if DA has a "concurrent" function in the striatum in driving tone-responses, how can one explain stimulus selectivity in the striatum ? Could DA acts as a permissive factor to "gate" striatal tone responses driven by auditory cortex?

While my comments above do not diminish the overall merit of the manuscript, I think it would be desirable to interpret the findings more carefully.

2) I think it is good practice to include a "drug-only" control, when doing chemogenetic inhibition experiments. This control can demonstrate that CNO alone is not responsible for the observed difference in tone-responses. This control seems to be missing currently.

Minor comments:

- Figure 3b is very unclear. If these are representative images, I am a bit concerned about the data quality obtained with this method.

-There are surprisingly few response recorded in both VEH and CNO sessions. Of 63 tone responsive (15,95% Figure 2) only slightly more than 25 are recorded in both CNO and VEH conditions. Is this due to the fact that the "sound cloud" evokes variable responses? Or is this related to technical issues with tracking neurons over two days (also see above)?

- The statement on p.5 line 12 ff is too vague and not really supported by the references. Please rephrase.

- p.7 line 18 typo "An SNC" should be "A SNC"

- p. 14 line 24 "In addition, silencing the auditory nigrostriatal pathway suppressed striatal sound ...": This is not correct, the authors do not show responses in the striatum were suppressed, they find that responses were reduced.

- There are different number of data points shown on different subplots (Fig. 2J – around 40, Fig.2L – 16). I don't find this explained in the text. Also, the methods and criteria for ROI tracking are not really described in the Methods, there is just reference to some Matlab package.

We thank the reviewers and editor for their careful reading and constructive remarks. We have taken the suggestions and revised the manuscript accordingly. Please find below a detailed point-by-point response to all comments (reviewer's comments in grey and italic, our responses in black).

Reviewer #1 (Remarks to the Author):

Chen et al. investigated the tail portion of the striatum (TS) in an auditory discrimination task using a two-alternative forced choice paradigm. Optogenetic inhibition of either the cell bodies or the terminals of the midbrain dopaminergic projection to the tail of striatum impaired task performance. Next, the Authors found by in vivo Ca imaging that a portion (about 15%) of TS neurons respond to sound. Surprisingly, sound responses were greater when the tones did not convey sensory evidence as compared to easy trials. Chemogenetic inhibition of dopaminergic neurons abolished these tone responses. In agreement, DA release in the TS and dopaminergic cell bodies in the SNc showed a similar activation pattern. Finally, D1 MSNs in TS were also shown to be tone-responsive and necessary for task performance. The TS represents a new hot topic and the Authors approached it from an unusual angle of perceptual decision making, making this an exciting and important study. The questions were addressed by multiple and adequate methods with many important control experiments; thus, the experimental design was thorough and appropriate. I have some reservations about data analysis and presentation: I believe the data could be better analyzed to embed the results in the context of the newly emerging TS field as well as the long-established dopamine field, thus engaging a broad audience. I think my points can be addressed by more in-depth analysis of the data already collected, after which this paper could represent an important advance in our understanding of the dopaminergic control of striatal circuits. Major points

1. The Authors characterize tone responsiveness of TS neurons. What I mostly miss from their analysis is a better embedding of their results into the vast literature of DA control over striatal functions, as well as into the newly emerging literature of TS.

We thank the reviewer for pointing out this shortcoming. In response to the detailed concerns and suggestions, we added additional analyses and clarifications in the text accordingly.

1a. Tone-responsive neurons constitute only 15% of TS. Do the other TS cells display different task-related activity, or no task-related activity at all? Is this proportion considered high compared to other parts of the striatum?

To respond to this concern, we revisited all the related recording data. We found that the proportion of tone-responsive neurons varied across animals (from 7% to 36%). The proportion was largely correlated with the field-of-view based on post hoc validation. The more posterior striatum in the field of view, the higher portion of tone-responsive neurons we observed. Indeed, TS has higher tone-responsive neurons than other parts of the striatum. We note that we didn't observe significant tone responses in the dorsomedial striatum in mice using the same recording paradigm. We added this new information in the revised text (page 8, lines 24-26).

As the reviewer suggested, we also revisited TS neuronal responses to other task events and found specific activities towards outcomes (reward or no reward). The neurons responding to the no-reward outcome may potentially represent the RPE activity. These neurons do not overlap with the tone-responsive neurons. We should note that we found some neurons responding to the white noise punished sound in early-withdraw trials, which partially overlapped with the tone-responsive neurons. The plot (**R1Fig.1**) below summarizes the overall quantification as well as individual animals' breakdown. We agreed with the reviewer that this is an interesting line of analyses. We updated **Figure 2C** and text (page 9 lines 1-4) in the manuscript accordingly to include the new analyses.

R1Fig. 1. Proportion of event-responsive neurons in the auditory striatum.

(A) Neuronal ROIs that are responsive to task events from all mice. **(B)** Individual pie charts for each mouse.

1b. Do TS neurons respond to reward? Does a part of TS neurons show RPE activity? The major finding of the negative correlation between DA input and evidence strength contradicts the RPE theory, which would predict stronger DA responses for cue stimuli of stronger reward-predictive strength. Some analysis is presented in Figure S3H, but the grand averages do not provide a clear picture. Can the larger DA responses at zero evidence be related to a larger probability of error and consequential time out, as some form of ‘threat prediction’ (see relevant papers of the Uchida lab)? In Fig. 3L, how where the ‘reward prediction error’ neurons defined? Tone responsive is supposed to be 32% according to the label, but that is at odds with the pie chart, where it looks very close to 25%. What does ‘responded to varying water reward levels’ mean? I was surprised to read this, as I could not find previous mentions of different reward sizes. This seems important in a task where neurons potentially related to RPE are probed, so this should be clarified. Were ‘tone responsive’ and RPE neurons non-overlapping? This is unclear, as I would expect RPE neurons to be tone-responsive as well in this task. Does this mean that tone responsive neurons are not feedback responsive or just not correlated with the reward size? What are the neurons in Fig. 3K? All recorded, ‘tone responsive’, or ‘tone-responsive in a difficulty dependent manner’? How do reward responses look as a function of evidence strength in the different subpopulations? How do different cells respond in error trials? Does this depend on evidence strength? This could be interesting for both SNc responses and DA sensor data.

This is a set of very interesting questions. We have revisited the data on this section, added new description and analyses, and corrected the mismatched pie. The details are as below.

As we presented in the response to 1a (**R1Fig. 1**), we found that some TS neurons respond to reward and no reward outcome, and the latter could be potential RPE activity. We revisited the original data and corrected the mismatched pie chart due to have used the incomplete data set for plotting in the original **Figure 3L**.

In this manuscript, the “reward prediction error” neurons refer to the SNc neurons responding differently to the reward sizes. This was conducted in a separate set of recordings, in which we vary the reward water sizes (we

added the description in **Result** page 13 lines 21-25 and **Method**, page 22 lines 1-3: “In a subset of recordings, we probed how SNc neuronal activity changes as a function of reward sizes. In these recordings, we manipulated the water reward sizes (2.5 μ l, 5 μ l, or 10 μ l) presented to mice that were well-trained in the auditory task with fixed water reward size (2.5 μ l).” Note that these neurons are not overlapped with the tone-responsive neurons in SNc. All the other analyses were from data collected in the tasks with fixed water size, and from the tone-responsive neurons (including **Figure 3K**).

Based on the RPE theory DA activity would be stronger for stimuli with higher evidence strength (± 1); or DA activity would be strongest suppressed for stimuli with higher possibility of aversive outcome – time-out (0 evidence strength). As we showed in **Figure 3D**, we didn’t find any suppression of DA towards the tone stimuli, suggesting the cues are less likely associated with the timeout. In **Figure 3F** we showed that there is no difference in striatal DA activity towards rewards with different prediction levels (evidence strength) and found the same from the error trials (**R1Fig. 2A**). Moreover, we found that in SNc the tone-responsive neurons do not respond to rewards, and the activities of reward-responsive neurons do not correlate with evidence strength of the cues (**R1Fig. 2B**).

R1Fig. 2. Dopamine sensor and SNc neuronal responses to outcome in error trials across evidence strength. (A) DA sensor recordings in the auditory striatum aligned to Choice-in (the timepoint of poking into the choice ports) during error trials. (n = 5 animals, 10 sessions). **(B)** Tone-responsive SNc neuronal activity aligned to Choice-in during error trials. (n=6 neurons, 8 sessions).

1c. Do TS and SNc DA neurons respond to the sound punishment for early withdrawal? How do responding neurons overlap with those categorized as tone responsive? Is there any activity related to the time-out punishments? The tones used here were 60 dB, which are borderline aversive. Do the Authors have evidence that they were not perceived as aversive, and the tone responses were not related to sound intensity? Especially given that tone-evoked dopamine sensor responses in the passive task seemed to be around the same magnitude as in the active version of the task, which is unlike most cue-evoked DA responses that emerge through the training. These questions seem important in light of data from the Uchida lab, which characterizes the system at study as ‘threat prediction error’ representation. A more thorough discussion of this literature and its relation to the present findings could help a lot in interpreting the results.

This is a great question. In response to this question, in the revision, we included new analyses (**Fig. 2C** and **Fig. 3L**, also copied below) to show that a few TS and SNc DA neurons respond to the white noise for early withdrawal. We should point out that they only partially overlap with the tone-responsive neurons --- the majority of tone-responsive neurons whose activities correlated with the evidence strength are not responsive to the white noise.

Furthermore, there are neurons that responded to the “no reward” outcome (timeout trials), but they are distinct from the tone-responsive neurons and their activities are not correlated with the evidence strength (R1Fig. 2B). Regarding the defensive behaviors, from both naïve and well-trained mice, we did not observe any noticeable avoidance or defensive behaviors towards the 60 dB sound intensity. Thus, we think that these cues are likely not aversive to the mice, particularly to the well-trained mice that had been in this daily training condition for more than a month. However, in a set of separate recordings, we did find stronger striatal DA signals evoked by tones at the beginning of the training session, which is consistent with Uchida Lab’s findings of DA activity towards novel cues. For the recordings in well-trained mice that are used in the analyses in this manuscript, the tone cues are no longer novel, and the DA responses are relatively stable. Altogether, these additional analyses together with the ones included in the original manuscript, suggest that the tone-evoked striatal DA activity observed in our study is related to the auditory discrimination rather than the RPE, likely not to the “threat prediction error” either. We felt that different populations of SNc neurons and TS neurons may mediate these functions differentially. We added a paragraph in discussion to address these concerns (page 17, lines 12-26, page 18 lines 1-4).

R1Fig. 3. Proportion of TS and TS-projecting SNc responsive neurons. (A) Proportion of TS neurons responsive to tones only, white noise and tones, white noise only, outcomes (reward in correct trials, no reward in error trials). “Other” refers to neurons showing no clear response toward any task events. **(B)** The same proportion analyses for the TS-projecting SNc neurons.

2. The Authors showed D1 MSNs may be necessary. It should be emphasized that this does not rule out the potential role of D2 MSNs too strongly. In this regard, in Fig.2C, were the counts normalized to relative density of D1 and D2 MSNs in TS? That is, is this truly preferential targeting? Relatedly, if I’m not mistaken, since all TS had 15% tone responsive population, and so had D1 MSNs, then necessarily D2 MSNs have the same 15% tone responsive subpopulation.

We agree with the reviewer. We have performed a new set of experiments and added new results from D2 manipulations. As shown in the revised **Figure 4F&G** (also briefly showed below), we found that optogenetic silencing D2-MSN or pharmacological inhibiting D2 receptors have little effect on mice’s task performance. We also added the corresponding results, discussion, and method on page 15 lines 1-17 (results), and page 18 lines 19-26 page 19 lines 1-4 (discussion), and page 27 lines 16-21 (method).

R1Fig. 4. Modulation of the auditory D2R Pathway does not significantly impact auditory discrimination.

(A) Left, schematic for bilateral optogenetic inhibition of the D2-MSNs through ArchT expression in A2a-Cre mice. Right, psychometric performance during auditory stimulus–based silencing comparing optogenetic light vs. masking light trials. Data per evidence strength are presented as the mean \pm SEM (n = 4 mice across 10 sessions). Insert: comparison of corresponding regression slopes for individual sessions. Bars represent the mean and dots represent the individual sessions (ns, $p > 0.05$, two-sided Wilcoxon rank-sum test). (B) Left, schematic for bilateral drug microinfusion of D2R antagonist Sulpiride and representative histology. Scale bar is 1000 μ m. Right, psychometric performance comparing saline vs. D2R antagonist sessions. Data per evidence strength are presented as the mean \pm SEM (n = 4 mice across 10 sessions). Insert: comparison of corresponding regression slopes for individual sessions. Bars represent the mean and dots represent the individual sessions (ns, $p > 0.05$, two-sided Wilcoxon rank-sum test).

3. The Authors make a point in the discussion that their results contradict a role of the TS DA input in learning. However, mice were overtrained and learning was not directly tested. Therefore, I think this claim should be removed, since in the present form, it is not implicated by the data.

We agreed with the reviewer and removed the corresponding discussion.

4. The Authors rightly realize that their results may be linked to confidence representations. They conduct an additional experiment; however, this is relegated to the Discussion section and remains quite obscure. First, the corresponding paragraph in page 17 is unclear. What question was addressed and how, what ‘discrepancy’ remained unresolved (line 7-8)? I think this experiment could be better presented in the Results, with some additional analyses that could shed some more light on the nature of the neuronal representations. Specifically, evidence strength, reaction time and peak DA responses were all correlated. Which of these three associations were stronger? Can computing partial correlations or mutual information help understand the relationship of these variables?

We agree with the reviewer’s comments and suggestions. We generated a separated supplementary figure to present the correlations among evidence strength, reaction time, and tone-evoked striatal dopamine response (**Supplementary Fig. 4**). The stronger correlation of tonal striatal dopamine response with evidence strength than with reaction time (**Supplementary Fig. 4D&F**) suggest that this dopamine activity may be closer to the auditory perception than decision making *per se*. We added the description in the result section (page 12 lines 15-26, page 13 lines 1-4) as well as the revised discussion (page 17 lines 2-7). The “discrepancy” refers to the opposite correlation between dopamine signals and task difficulty observed in previous studies. We revised the sentence to make it clear.

R1Fig. 5. Correlations among evidence strength, striatal dopamine activity, and reaction time. (A) Correlation plot and R^2 between evidence strength and the same trial tone-evoked DA sensor peak responses. (B) Correlation plot and R^2 between evidence strength and same-trial reaction time. (C) Correlation plot and R^2 between reaction time and the same trial tone-evoked DA sensor peak responses. Blue dots are individual trials, and the red line is the correlation regression. n = 3 mice, 6 sessions, 191 trials.

Minor points:

1. Chemogenetic inhibition in Fig.2 seems to largely abolish tone responses. Therefore, it seems rather trivial that differences related to task difficulty (evidence strength) also disappear. Maybe this part could be rephrased to make this clearer.

We thank the reviewer for pointing out this confusion. We replaced the representative tracings in **Figure 2 F-I** to show that CNO dampened the tone responses but not completely abolished which is also reflected in **Figure 2K**. The representative traces are also shown below:

R1Fig. 6. Representative trace responses of individual neuronal ROI responses towards different tonal types during both vehicle and CNO sessions (placed into Fig. 2F-I). (A-D) Averaged tone trace response of example neuronal ROIs in vehicle (red) and CNO (green) sessions for high frequency (A), low frequency (B), 1 evidence strength (C), and 0 evidence strength (D) stimuli.

2. It is hard to see the point of Figure S3F and S3G for those trials in which the stimuli were of zero-evidence, as both ports had equal chance of being the correct port. On the other hand, I would be curious whether this figure looks the same for the other trials.

We agree with the reviewer and added plots for other evidence strength to (**Figure S3F**, also attached below) showing that the tone-evoked striatal DA activity does not predict the correct rate of the decisions. Due to the relatively short recording durations for each session (12 min), not every session has trials for all evidence strengths. Therefore, we only include sessions with trials for a given evidence strength. For evidence strength 1 and 0.33, there are a lot more correct trials than error trials. To better compare the tonal response and variation, we balanced the trial numbers by randomly selecting the same numbers of correct trials from all correct trials.

R1Fig. 7. Tone-evoked DA responses between correct and error trials. Left, tonal responses for evidence strength 1.0 ($n = 3$ animals; 12 sessions, 82 error trials and 82 trials randomly selected from 950 correct trials). Middle, tonal responses for 0.33 evidence strength ($n = 3$ animals, 8 sessions, 281 error trials and 281 trials randomly selected from 462 correct trials). Right, tonal responses for 0 evidence strength ($n = 3$ animals, 10 sessions, 309 for error and 309 for correct trials).

3. p. 4. line 11 'dorsal that' should probably read 'dorsal striatum that'; p 13 line 16 'zone' not 'zones'
We have corrected them.

4. Fig. S3L, S5B-D are barely visible.
We have revised them to make brain contours darker in color.

5. Fig. 3F - What is choice-in onset? Please explain or find a better label.
Choice in onset refers to the time point at which the mice poked into the side ports. We revised the event timelines in **Figure 1 E&H** to introduce it. We also revised the **Figure 3F** legend to clarify it.

6. Fig. 3I - Yellow trace is not visible.
We have revised it to blue color.

7. Fig. S4 shows an example neuron. Adding averages across cells could make this figure more convincing.
The reviewer may refer to **Supplementary Figure 4B** (new **Supplementary Figure 5B**). We added a quantification plot to show individual neurons as well as averaged responses as a function of water size presented.

R1Fig. 8. Reward-responsive SNc neuronal activity at different reward sizes. Left, one example SNc neuronal activity. Right, quantification of the six SNc neurons tested in different reward sizes. Bars are mean and error bars are SEM.

8. Fig. S4B: x axis 3s tick label is misplaced.
We have revised it.

9. Fig. S5 title should be rephrased
We have revised it to "Additional anatomical and behavioral characterization".

10. Discussion: 'Our results showed a positive correlation between striatal dopamine release and the evidence strength of presented stimuli (Fig. 3).' Did you mean negative correlation?
We have corrected it.

I believe that openness and transparency can increase the fairness of peer review. Therefore, I decided to sign my reviews.

We appreciate Dr. Balazs Hangya's critiques, comments and suggestions.

Reviewer #2 (Remarks to the Author):

Chen et al. have investigated a nigrostriatal dopamine pathway and its role in regulating auditory discriminatory behaviors. The authors demonstrated that the optogenetic inhibition of nigrostriatal dopaminergic projections specifically impairs the mouse choice performance but not a movement in an auditory frequency discrimination task. The authors also test the hypothesis that increasing the task difficulty influences the dopamine dynamics in the auditory striatum. To test their hypothesis, they use a combination of the microendoscopic approach and the dopamine fluorescence sensor (the G-protein-coupled receptor activation–based series DA2m).

1. The authors indicated that they delivered optical stimulation during both pre-tone and tone phases to achieve sufficient silencing during tone presentation. Can the authors provide further evidence that the optogenetic suppression produced a silencing of the neurons in the SNc?

We thank the reviewer for pointing out this missing information. We added a new data plot in **Supplementary Figure 1B** (copied below) to show that the optogenetic stimulation suppressed SNc neuronal activity. In brief, AAV-FLEX-ArchT-GFP was injected into SNc of DAT-Cre mice, and brain slices were acutely prepared for patch clamp recordings. Both cell-attached and whole-cell recordings showed that LED light pulses silenced the ArchT-expressing SNc neurons. Corresponding descriptions are added in the result (page 7 lines 1-3) and method (page 26 lines 19-25) sections.

R2Fig. 1. Optical stimulation of ArchT silences tonic firing in SNc neurons. (A) Left, example cell attached voltage recording from an ArchT-GFP -positive SNc neuron showing tonic firing at rest and transient silencing in response to a LED pulse (blue box). Right, quantification of firing suppression from all recorded neurons (n =5). (B) Left, example whole cell current clamp recording from an ArchT-GFP -positive SNc neuron showing a robust hyperpolarization in response to a LED pulse (blue line). Right, quantification of hyperpolarization amplitude from all 5 recorded neurons.

2. The authors should provide more results about when the auditory nigrostriatal neurons are active during the behavioral task (for example, extracellular recording). This will give the base to manipulate the activity (activation/silencing) of the SNc neurons during the behavioral task.

We agreed with the reviewer and extended our analyses on SNc neuronal activities to different task events. The result is updated in **Figure 3L** (also shown below), in which we showed that for the neurons showing specific

activities during the task. Importantly, majority of them responded to the tones. This is consistent with our effective optogenetic manipulation time windows in **Figures 1E-I, 4D, 4F**.

R2Fig. 2. SNc neuronal responses to task variables. (A) One example SNc neuron responding to tones. (B) Pie chart depicting the distribution of responsive neuronal ROIs. ‘Tone-responsive’ neuronal ROIs refer to ROIs with significantly ($p < 0.05$) higher response profiles following different task variables: tones, white noise (in early withdraw trials), reward (in correct trials), or no reward (in error trials) are quantified. cue presentation. ‘Reward prediction error’ neuronal ROIs refer to those with significantly higher response profiles following the receipt of larger water volume rewards compared with smaller water volume rewards ($p < 0.05$). ‘Other’ refers to neurons with no significant responses to task variables. Fluctuations in dopamine sensor activity and neuronal activity are both denoted as Z-Score $\Delta F/F$.

3. The authors performed an experiment to verify that this phasic response represents physiological changes in dopamine. They injected the D2R-specific antagonist eticlopride and observed blunted tone-evoked DA2m responses in mice during cued tasks. In addition, the external application of another D2R-specific antagonist, haloperidol, diminished the increase in DA2m fluorescence evoked by SNc axon terminal activation using channelrhodopsin-2 in acute brain slices (Supplementary Fig. 3D). Can the authors provide more evidence in the results section of how this experiment was performed?

We agree with the reviewer for including more details for these experiments. We have added the descriptions to both the result and method sections (page 11 lines 5-15, page 25 lines 22-26, page 26 lines 1-16, page 27 lines 1-3). Also copied as below:

Result section

To verify that this phasic response represents physiological changes in dopamine, we took advantage of the structural quenching of DA2m by D2 dopamine receptor (D2R)-specific antagonists (Sun, Zhou et al. 2020). We first adopted a reported *in vivo* protocol ((Sun, Zhou et al. 2020), also see in **Method**) and found that intraperitoneal (i.p.) injection of D2R-specific antagonist eticlopride (1.0 mg/kg) blunted tone-evoked DA2m responses in mice performing the task (**Supplementary Fig. 3B&C**, $p < 0.0001$). We next used acute brain slices to further verify the DA2m in our system. The external application of another D2R-specific antagonist, haloperidol (10 μ M, (Sun, Zeng et al. 2018)), diminished the increase in DA2m fluorescence evoked by SNc axon terminal activation using channelrhodopsin-2 (ChR2, (Prager, Dorman et al. 2020))(Supplementary Fig. 3D). These data suggest that the tone-evoked increase in DA2m fluorescence observed in **Fig. 3B&C** is due to local changes in dopamine concentrations during task performance.

Method Section

In vivo validation of DA sensor imaging

It has been previously validated that *in vivo* i.p. injection of eticlopride can blunt nigrostriatal-induced DA fluctuations (Sun, Zhou et al. 2020). Here, we used a similar method and performed i.p. injections of either saline or eticlopride dissolved in saline (1.0 mg/kg, Cayman Chemicals, Ann Arbor MI) in well-trained mice 15 minutes

prior to placing mice in behavior rigs and recording setups. Recordings were performed for a continuous 8-minute period and tone-evoked responses were analyzed in the same manner as with all other in-task recordings.

Ex vivo validation of DA sensor imaging

To validate nigral control of DA levels and thus DA2m-based fluorescence, we employed an *ex vivo* approach described previously (Sun, Zeng et al. 2018). We infused AAV9-CAG-DA2m in the auditory striatum and AAV5-CAG-ChR2 in the SNc of wild type mice. 3-5-week after viral infusion, mice were anesthetized with a ketamine/xylazine mixture (100mg/kg and 7mg/kg, respectively) and underwent transcardial perfusion with ice-cold artificial cerebral spinal fluid (ACSF) consisting of 124mM NaCl, 3mM KCl, 1mM CaCl₂, 26mM NaHCO₃, 1mM NaH₂PO₄, 14mM glucose, and 1.5mM MgCl₂. 275µm para-sagittal slices were obtained using a vibratome (VT1000s; Leica Microsystems, Buffalo Grove, IL). The auditory striatum's location was determined based on the sagittal anatomic coordinates and expression of DA2m. Slices were subsequently placed in the recording chamber of a two-photon Ultima Laser Scanning Microscope System (Bruker Nano, Inc., Middleton, WI) and bathed with ACSF solution (containing 2mM CaCl₂ and 1mM MgCl₂) at room temperature for recording. To activate ChR2-infected SNc terminals in the auditory striatum, five 1 ms duration optical stimulations were delivered at 30 Hz using a single photon 473 nm laser tuned to a power of 3 mW (Coherent OBIS FP 473LX, Coherent, Inc., Santa Clara, CA). Spiral line scans (920 nm, 21.2 ms with 0.0776 µm 2 pixels and 10µs dwell) were performed to measure DA2m fluorescent changes. A high-speed shutter was used to block cross-signaling between optogenetic stimulation and DA2m fluorescence measurements. Slices were either bathed in ACSF or ACSF with dissolved Haloperidol (10 µM; Sigma Aldrich).

4. *Chen et al. study indicated that D1-MSNs in the auditory striatum respond to tone stimulation during the auditory discrimination task, and the inhibition of D1-MSN activity or D1 receptors in the auditory striatum impairs auditory discrimination performance. Their experiment used an anterograde transsynaptic tracing approach to identify neuron targets of SNc projections in the auditory striatum. This experiment indicated that both tdTomato+ D1 and D2 neurons were transsynaptically labeled in the auditory striatum. For this reason, it is crucial for this study to demonstrate the role of D2-MSN neurons such as examine whether tone-evoked responses occur in the D2-MSNs during the auditory task and determine whether tone-evoked D2-MSN activity is essential for auditory discrimination behavior.*

We agree with the reviewer on the necessity for including the complete set of D2-related tests. In the revised **Figure 4F&G** (also showed below), we added new results from D2 manipulations and showed that optogenetic silencing D2-MSN or pharmacological inhibiting D2 receptors have little effect on mice's task performance. Corresponding result, discussion and method are added on page 15 lines 1-17 (result), and page 18 lines 19-26 page 19 lines 1-4 (discussion), and page 27 lines 16-21 (method).

R2Fig. 3. Modulation of the D2R Pathway does not significantly impact auditory discrimination.

(A) Left, schematic for bilateral optogenetic inhibition of the D2-MSNs through ArchT expression in A2a-Cre mice. Right, psychometric performance during auditory stimulus-based silencing comparing optogenetic light vs. masking light trials. Data per evidence strength are presented as the mean ± SEM (n = 4 mice across 10 sessions).

Insert: comparison of corresponding regression slopes for individual sessions. Bars represent the mean and dots represent the individual sessions (ns, $p > 0.05$, two-sided Wilcoxon rank-sum test). **(B)** Left, schematic for bilateral drug microinfusion of D2R antagonist Sulpiride and representative histology. Scale bar is 1000 μm . Right, psychometric performance comparing saline vs. D2R antagonist sessions. Data per evidence strength are presented as the mean \pm SEM ($n = 4$ mice across 10 sessions). Insert: comparison of corresponding regression slopes for individual sessions. Bars represent the mean and dots represent the individual sessions (ns, $p > 0.05$, two-sided Wilcoxon rank-sum test).

Reviewer #3 (Remarks to the Author):

In the manuscript entitled “Nigrostriatal dopamine pathway regulates auditory discrimination behaviour” by Chen et al., the authors use a forced choice tone discrimination task in mice to investigate the role of dopamine projections to the tail of the striatum, referred to “auditory striatum” by the authors. They find that optogenetic inhibition of the tail of the striatum during the choice period impairs discrimination of tones. They then demonstrate that chemogenetic inhibition of striatonigral dopamine projections reduces tone responses in the tail of the striatum. Direct imaging of DA activity at the soma and release at the terminals in the tail of the striatum, revealed DA responses during tone presentation. Finally, they authors show that behavioral performance involves D1- but not D2-receptor neurons.

The manuscript includes an impressive series of experiments, using state-of-the-art tools to address an interesting question. Building on previous findings that D1-receptor neurons in the tail of the striatum are involved in performance in the perceptual choice task, the new experiments uncover a function of dopamine in mediating striatal tone-related neural activity and behavioural performance.

We thank the encouraging comments.

Major comments:

1) The author’s interpretation of the observed neural responses raises several questions for me. For the results shown in Figure 2, the authors seem to favour the interpretation that neural responses to tones in the tail of the striatum are related to “general perception” (p.17 l.9). I find this conclusion a bit difficult to follow. While I don’t question that neurons are activated by sounds in the task, it is not clear from the analysis done what these neurons actually respond to. What is the response selectivity of neurons to different sounds (e.g. Guo et al. 2018) ? Are there auditory responses in the tail of the striatum of untrained animals?

Similarly, the interpretation of DA responses shown in Figure 3 is not entirely clear. What are DA neurons encoding ? DA response to sensory stimuli that predict a possible reward are generally consistent with RPE. Maybe the correlation of DA and evidence could be related to the higher stimulus complexity of the low-evidence sound cloud or to (cognitive) effort ? Also, if DA has a “concurrent” function in the striatum in driving tone-responses, how can one explain stimulus selectivity in the striatum ? Could DA acts as a permissive factor to “gate” striatal tone responses driven by auditory cortex?

While my comments above do not diminish the overall merit of the manuscript, I think it would be desirable to interpret the findings more carefully.

We thank the reviewer for the fair and encouraging comments and pointing out these confusions. To clarify these confusions and include all the suggestions, we revised our text and figures accordingly.

We found tone-evoked striatal responses in both naïve and well-trained mice, from both current study and earlier works (Xiong, Znamenskiy et al. 2015). The plot below shows the results from recordings in two naïve mice (R3Fig. 1A). The auditory striatal neurons do have tone frequency selectivity, which is relayed from the auditory cortex (Chen, Wang et al. 2019). A set of examples showing neurons preferentially sensitive to low or high frequency tones are shown below (R3Fig. 1B). Furthermore, these neurons’ responses to tones remain the same regardless the mice’ movement decisions. Below (R3Fig. 1C) is an example of tone-evoked responses from high-frequency preferred neurons in correct trials (mice moved to rightward) and error trials (mice moved to leftward). In our task, low frequencies were randomly drawn from 5-10 KHz, and high frequencies from 20-40 KHz. Although each session had many low/high frequency trials based on category, not many trials have the identical low/high frequencies. Furthermore, due to the short recording duration (12 min each session), we did not always get both correct and error trials from the same stimuli. Thus, we only showed here 8 neurons with recorded responses towards the same stimuli in both correct and error trials. Together, we think the tone-evoked striatal response observed in our study is mainly auditory perception rather than decision or movement. The revised text and figures are on page 9 lines 4-8 and **Supplementary Figure 2B-D.**

Based on RPE theory, the stimuli associated with higher probability of reward will activate stronger DA signals, and unexpected reward (or aversive outcome) will activate (reduce) the DA signals. However, in our study we

found stimuli with 0 evidence induced the strongest DA signals, and the tone-responsive DA neurons do not respond to unexpected outcomes (neurons responded to no reward or larger reward than usual are not overlapped with these tone-responsive neurons, **Figure 3L** and **Supplementary Figure 4B**). Thus, we think these neurons do not encode RPE. In our stimuli, all the tone clouds contain the same numbers of the tones and the same stream pattern, the difference is at the tone frequencies. Therefore, the complexity is the same for different stimuli across trials. The auditory striatal neurons respond to tones with frequency preferences, but response of striatal DA does not have frequency preferences. The striatal DA response to tones with -1 evidence strength (low frequency tones) is about the same as 1 evidence strength (high frequency tones), and same for ± 0.33 and ± 0.67 evidence strength (**R3Fig. 1D**). As suggested by the reviewer DA signals may correlate with the difficulty/ambiguity of the stimuli (cognitive effort), and DA may function as a modulator to gate striatal tone responses. We included this in the revised discussion (page18 lines 5-7).

R3Fig. 1. Auditory striatal tonal response properties. (A) Striatal tone responses in naive mice. Left, tone-evoked response from one example neuron. Right, quantification from all neurons recorded in naive mice. (B) Example auditory striatal neurons with tone frequency preferences: high-frequency preferred neurons (left) and low-frequency preferred neurons (right). (C) Example neurons showing response towards the same stimuli in correct and error trials. Left, one example neuron. Right, quantification from the eight neurons (6 sessions, 142 trials). (D) Striatal DA response to tone cues across different evidence strength.

2) I think it is good practice to include a “drug-only” control, when doing chemogenetic inhibition experiments. This control can demonstrate that CNO alone is not responsible for the observed difference in tone-responses. This control seems to be missing currently.

We agreed with the reviewer. In the revised manuscript, **Supplementary Figure 2E&F** (also copied below) showed that CNO infusion to mice expressing mCherry only (no hm4Di) in SNc has no significant effect on striatal tone-evoked responses.

R3Fig. 2. CNO has no effect on striatal tone response in control mice expressing mCherry. (A) Left, schematic for mCherry control for recording the effects of clozapine N-oxide (CNO) delivery on auditory striatal activity. Right, tone-responsive auditory striatal activity during vehicle and CNO treatment sessions. (B) Effects of CNO delivery on auditory striatal neuronal calcium event rate activity ($p = 0.67$, paired t-test).

Minor comments:

- Figure 3b is very unclear. If these are representative images, I am a bit concerned about the data quality obtained with this method.

We thank the reviewer for pointing out this confusion. In the revised figure legend in **Figure 3B** and text page 10 lines 23-25 we clarify that this is a snapshot of DA sensor live imaging for detecting whole-field changes in the auditory striatum. All the DA activity analyses in **Figure 3A-G** are looking at the overall fluorescent intensity changes of DA sensor in the auditory striatum under the imaging field. The DA2m sensor is a membranous protein that ubiquitously express throughout the neurons. It has a challenge to obtain single cell resolution. We included a short video to demonstrate the recordings.

-There are surprisingly few response recorded in both VEH and CNO sessions. Of 63 tone responsive (15,95% Figure 2) only slightly more than 25 are recorded in both CNO and VEH conditions. Is this due to the fact that the “sound cloud” evokes variable responses? Or is this related to technical issues with tracking neurons over two days (also see above)?

We thank the reviewer for bringing this up. Indeed, as the reviewer mentioned, the few responsive neurons recorded in these sessions is due to the technical issue of tracking neurons across days/sessions.

- The statement on p.5 line 12 ff is too vague and not really supported by the references. Please rephrase.

We revised the statement as suggested by the reviewer. “Dopamine dysfunction has been linked to decision-making deficits in a variety of mental disorders such as Parkinson’s disease.”

- p.7 line 18 typo “An SNC” should be “A SNC”

We have corrected this.

- p. 14 line 24 "In addition, silencing the auditory nigrostriatal pathway suppressed striatal sound ...": This is not correct, the authors do not show responses in the striatum were suppressed, they find that responses were reduced. We agree and have replaced the word "suppressed" to "reduced".

- There are different number of data points shown on different subplots (Fig. 2J – around 40, Fig.2L – 16). I don't find this explained in the text. Also, the methods and criteria for ROI tracking are not really described in the Methods, there is just reference to some Matlab package.

To clarify this confusion, we revised the text in result section (page 10 lines 2-3) to clarify the different data points in **Fig 2J & L**. This is due to the technical limitation that we only be able to track 16 out of 42 neurons across Veh and CNO sessions.

We added detailed method for ROI tracking in the method section (page 25 lines 7-11), also copied below: "Specifically, we used a probabilistic cellular registration algorithm which estimates the probability of correct registration of a spatial footprint across multiple imaging sessions. A pair of cells is considered to have the same identity if the calculated probability is $P > 0.5$ with a centroid distance of 5 μm . Spatial components using this method were subsequently inspected for cellular shape and similarity of calcium transient dynamics across behavioral sessions."

References

Chen, L., X. Wang, S. Ge and Q. Xiong (2019). "Medial geniculate body and primary auditory cortex differentially contribute to striatal sound representations." Nat Commun **10**(1): 418.

Prager, E. M., D. B. Dorman, Z. B. Hobel, J. M. Malgady, K. T. Blackwell and J. L. Plotkin (2020). "Dopamine Oppositely Modulates State Transitions in Striosome and Matrix Direct Pathway Striatal Spiny Neurons." Neuron **108**(6): 1091-1102 e1095.

Sun, F., J. Zeng, M. Jing, J. Zhou, J. Feng, S. F. Owen, Y. Luo, F. Li, H. Wang, T. Yamaguchi, Z. Yong, Y. Gao, W. Peng, L. Wang, S. Zhang, J. Du, D. Lin, M. Xu, A. C. Kreitzer, G. Cui and Y. Li (2018). "A Genetically Encoded Fluorescent Sensor Enables Rapid and Specific Detection of Dopamine in Flies, Fish, and Mice." Cell **174**(2): 481-496 e419.

Sun, F., J. Zhou, B. Dai, T. Qian, J. Zeng, X. Li, Y. Zhuo, Y. Zhang, Y. Wang, C. Qian, K. Tan, J. Feng, H. Dong, D. Lin, G. Cui and Y. Li (2020). "Next-generation GRAB sensors for monitoring dopaminergic activity in vivo." Nat Methods **17**(11): 1156-1166.

Xiong, Q., P. Znamenskiy and A. M. Zador (2015). "Selective corticostriatal plasticity during acquisition of an auditory discrimination task." Nature **521**(7552): 348-351.

REVIEWER COMMENTS

Reviewer #1 (Remarks to the Author):

The Authors responded to my comments and the manuscript has significantly improved. I have no further comments and congratulate the Authors on this impressive set of experiments that will further our understanding of the diverse roles of dopaminergic projections.

I believe that openness and transparency can increase the fairness of peer review. Therefore, I decided to sign my reviews.
Balazs Hangya

Reviewer #3 (Remarks to the Author):

In their revised manuscripts the authors have addressed the followings concern appropriately

1) Additional information about the response properties of “auditory” neurons in the striatum is provided. The authors also clarified my question related to the complexity of the stimuli.

Please address the following concerns:

2) I think the CNO-only control is troublesome. The authors only show one read-out (event rate) which demonstrates that event rates similar between veh. and CNO in animals injected with fluorescent reporter. However, event rates are much lower than in hm4Di-expressing animals. No information about the number of control animals can be found in the ms. The appropriate statistical comparison would be between hm4Di/control groups injected with CNO. If there are doubts about these experiments, the authors should remove it from the manuscript. The manuscript can stand without it.

3) It is important to highlight that the authors collected extracellular DA signals in the striatum and not somatic DA responses. The dynamics of these two signals can be very different. Hence, the data does not directly speaks against RPE coding of DA neurons. Please add a sentence or two highlighting what physical measurement was collected (extracellular DA vs somatic Ca²⁺).

Minor comments:

- line 3: Please avoid such general claims, when the manuscript does not provide any specific evidence in favor of the claim.

We thank all the reviewers for their remarks and thank the reviewer #3 for his/her further comments on our revision. Here, we have accepted the suggestions with additional data and analyses and revised the manuscript accordingly. Please find below a detailed point-by-point response to all the comments from the reviewer #3 (reviewer's comments in grey and italic, our responses in black).

Reviewer #3:

In their revised manuscripts the authors have addressed the followings concern appropriately
1) Additional information about the response properties of "auditory" neurons in the striatum is provided. The authors also clarified my question related to the complexity of the stimuli.

We thank the reviewer for their original comments and appreciate the encouraging feedback regarding our prior revision.

Please address the following concerns:

2) I think the CNO-only control is troublesome. The authors only show one read-out (event rate) which demonstrates that event rates similar between veh. and CNO in animals injected with fluorescent reporter. However, event rates are much lower than in hm4Di-expressing animals. No information about the number of control animals can be found in the ms. The appropriate statistical comparison would be between hm4Di/control groups injected with CNO. If there are doubts about these experiments, the authors should remove it from the manuscript. The manuscript can stand without it.

We thank the reviewer for pointing out the missing read-out, peak z-score analysis, which is used for other data sets. We also thank the reviewer for pointing out the big variation of event rates between mCherry and hM4Di groups, and the missing comparison between CNO groups. To address these concerns, we performed additional analyses and revised the figures accordingly. We also added the missing information of animal numbers in corresponding figure legend. Briefly summarized as below:

Calcium signal peak z-score is the main parameter used in our manuscript to quantify neuronal activity. In the revised **Figure 2E** (also showed below as **R2Figure 1**), we included neuron peak z-score quantifications of CNO vs vehicle results for both mCherry and hM4Di groups. The optic imaging method allowed us to track the same cohort of neurons across training sessions such as CNO and vehicle conditions. Because different neurons respond to the tones very differently, our first comparisons were done on individual neurons between CNO and vehicle conditions.

Importantly, as suggested by the reviewer, we next performed the comparison between mCherry and hM4Di groups in CNO condition and found them significantly different ($p < 0.05$, $n = 21$ neurons from 3 mCherry mice and $n = 21$ neurons from 3 hM4Di mice). We have more recorded neurons ($n = 27$ neurons) from hM4Di mice as shown in **Figure 2D**. To balance the numbers of neurons for comparison between mCherry and hM4Di groups, we randomly drew 21 neurons from the hM4Di group to match the total number in mCherry group.

Note that for the calcium event rate, indeed there were big variations across mice as identified by the reviewer. Considering this variation across experimental animals and that the rest of the manuscript did not use this parameter for any quantification and conclusion, we decided to focus on peak z-score analyses and remove the event rate analysis presentation.

R2Figure 1. CNO impact on striatal tonal responses. Neuronal peak Z-score tonal responses in hM4di- and mCherry-expressing mice. Peak trace response across registered neuronal ROIs for vehicle (red) and CNO (green) conditions. On the left in the bar plot is the data from hM4di-expressing mice (** $p < 0.01$, Mann–Whitney U test, 21 neurons in 3 mice). On the right is the data from mCherry-expressing mice (n.s., $p > 0.05$, Mann–Whitney U test, 21 neurons in 3 mice). A statistical test was used to compare the tone responses between the two cohorts in CNO condition (* $p < 0.05$, Mann–Whitney U test). Individual dots are individual neuronal ROIs registered for both vehicle and CNO sessions.

3) It is important to highlight that the authors collected extracellular DA signals in the striatum and not somatic DA responses. The dynamics of these two signals can be very different. Hence, the data does not directly speak against RPE coding of DA neurons. Please add a sentence or two highlighting what physical measurement was collected (extracellular DA vs somatic Ca²⁺).

We agree with the reviewer and added the following sentences to the discussion (pg. 18, lines 4-6):

“We note here, however, that our DA sensor measurements only allowed us to visualize extracellular dopamine fluctuations. It is possible that such fluctuations do not necessarily correspond to somatic downstream signaling.”

Minor comments:

- line 3: Please avoid such general claims, when the manuscript does not provide any specific evidence in favor of the claim.

“line 3 of the abstract: “Despite its high clinical importance, the dopaminergic modulation of sensory striatal neuronal activity and its behavioral influences remain unknown.”

Either the authors refer to something specific, or change the sentence. If I have not missed something majorly, the ms does not contribute much directly to understanding anything clinically important. While this does not reduce the importance or impact of the research at all, a phrase like the one above somehow implies it does. “

We agreed with the reviewer and have rephrased it as below:

“Despite its known connections to diverse neurological conditions, the dopaminergic modulation of sensory striatal neuronal activity and its behavioral influences remain unknown.”

REVIEWERS' COMMENTS

Reviewer #3 (Remarks to the Author):

The authors have adressed my concerns. Ready for publication! I also with to thank the authors for their patience in the review process.

Reviewer #3 (Remarks to the Author):

The authors have addressed my concerns. Ready for publication! I also wish to thank the authors for their patience in the review process.

We thank the reviewer for the suggestions in the whole review process.